# Emotion Regulation Strategies in Educational, Work and Sport Contexts: An Approach in Five Countries

**DOI:** 10.3390/ijerph20196865

**Published:** 2023-09-29

**Authors:** Silvia Cristina Da Costa Dutra, Xavier Oriol Granado, Darío Paéz-Rovira, Virginia Díaz, Claudia Carrasco-Dajer, Alicia Izquierdo

**Affiliations:** 1Psychology and Sociology Department, Faculty of Social and Human Sciences, University of Zaragoza, Teruel Campus, 44003 Teruel, Spain; aliciaig@unizar.es; 2Psychology Department, Faculty of Education and Psychology, University of Girona, 17004 Catalonia, Spain; xavier.oriol@udg.edu; 3Faculty of Education and Social Sciences, University Andres Bello, Santiago 7590924, Chile; dariopaez@hotmail.fr; 4Social Psychology Department, Faculty of Psychology, University of the Basque Country, 20018 Donostia/San Sebastian, Spain; virginia.diaz@ehu.eus; 5Department of Nursing Sciences, Faculty of Medicine, Catholic University of the Most Holy Conception, Concepción 4990541, Chile; ccarrasco@ucsc.cl

**Keywords:** adjustment, affect, emotion, regulation, well-being

## Abstract

One of the greatest challenges in the domain of emotional regulation is comprehending the functionality of strategies and their utilization in various social contexts. In this sense, this study analyzes differences in the use and efficacy of regulation strategies, particularly of interpersonal strategies like altruism, social support, negotiation, mediation, regulation, and rituals, in samples of workers (*N* = 687) and students (*N* = 959) from Brazil, Chile, Cuba, Spain, and Uruguay, and athletes (*N* =144) from Spain. Participants answered questions pertaining to measures of affect or emotional regulation (MARS and ERQ self-regulation scales and EROS heteroregulation), as well as questions of a wellbeing scale (PHI) and questions related to emotional creativity (ECI), humor styles (HSQ), and adjustment to stress. Athletes reported less emotional discharge, use of humor, and affection, and greater confrontation and use of rituals than students and workers. A congruent relationship was found between the use of functional strategies (like direct coping, distraction, reevaluation, and active physiological regulation) and adjustment to stress, well-being, and creativity. Seeking social support, negotiation, and, to an extent, altruism, confirmed their predicted adaptive character. Mediation and delegation did not confirm their predicted adaptive character. Rumination, social comparison, rituals, confrontation, and suppression were maladaptive for workers and students, but the first four strategies were functional for athletes, who display a higher self-control and a more team-oriented and competitive emotional culture. Finally, the results show that adaptive regulation strategies mediate the relationship between well-being and adjustment to stress.

## 1. Affect, Mood, and Emotions

Affect is defined as the subjective experience of the general emotional state, which can be positive/pleasant or negative/unpleasant [1]. Emotional episodes typically result in a mixture of emotions, while also reinforcing, creating, or relying on mood states [2]. Emotions shape individuals’ subjectivity through the process of symbolic interaction between identity, norms, and social values [3], and influence the behaviors that can be observed in different social contexts. Despite emotions being conceived by researchers from very different perspectives, there appear to be some key similarities among experts [4]. First, emotions involve moderately related changes in domains such as subjective experience, behavior, and physiological aspects. Second, they develop over time [5]. In this sense, the Process Model of Emotion Regulation proposed by [4] provides a framework for understanding how emotions are generated and regulated, and how different components of emotional responses interact with each other. According to this model, the sequence begins with the appraisal of a psychologically relevant situation. The appraisal process determines whether the situation is perceived as positive or negative, and whether it is relevant to the individual. 

Whether of one kind or another, these situations are e evaluated in terms of what they mean to the individual based on their goals [6]. Finally, the model suggests that emotions are important because of their relationship with prominent social beliefs [7] and ethical decisions [8,9]. Current research concludes that emotions and cognitive processes mutually influence each other and are co-determined by sociocultural processes [10,11,12]. In this sense, there is an important consensus that emotions are not just individual, internal experiences but are shaped by cultural norms, social contexts, and historical and political circumstances [13].

## 2. Affective Regulation

Emotional regulation is a process considered necessary for optimal personal and social adaptation [14]. Individuals who possess well-developed emotional competencies to regulate their affectivity report having more positive relationships with others, exhibit fewer antisocial behaviors, and are less prone to psychological disorders compared to those who exhibit lower levels or lack such competencies [15]. While affect regulation refers to the modification of moods, emotional regulation refers to individuals’ capacity to modify components of their emotional experience.

According to the Process Model of Emotion Regulation in [4], emotional regulation begins when a person chooses between possible situations based on their probable emotional impact. The model suggests that once an emotion is generated, it can be regulated through various cognitive and behavioral strategies which modify the appraisal, physiological arousal, and expression components of the emotional response. These strategies include the following: (1) the modification of the situation, which involves altering the situation to modify its emotional impact; (2) attentional deployment, which involves focusing attention on one aspect of the situation; (3) cognitive change, which involves modifying the meaning attributed to the situation, according to which the potential emotional meanings would lead to a new decision, resulting in the experience, behaviors, and physiological response tendencies that define the emotion; and, finally, (4) modulation of affective response refers to directly influencing the experience, behavior, or physiological components of the emotional response after the emotion has been generated.

### 2.1. Goals of Affect Regulation

Regulation is considered functional if adaptive goals are achieved [16]. Different motivations or goals for emotional regulation are important. First, arousal regulation (which increases when people are bored or decreases when they are upset) and, second, hedonic regulation, usually related to a decreased negative affect and improved positive affect, are two basic goals related to affect valence and activation. Third, instrumental or problem solution goals or task-related (or achievement outcomes) goals are important. Fourth, goals of social integration or goals with belonging-oriented outcomes are relevant in general and especially in the workplace. Other goals or motives are important, like maintaining a positive self-image and defending self-esteem and cognitive motives, or understanding emotions and affect [17,18]. Briefly, affect regulation can be characterized by the inclination to improve feelings and arousal, solve tasks successfully, defend self-esteem, and understand and control experience, as well as to maintain or improve social integration. 

### 2.2. Relational and Social Regulation Forms

Forms of regulation are important for all goals, but also particularly for social integration goals, which are those that are carried out through interactions with others. Here, not only the search for social support, widely studied, but also forms of regulation that involve mobilizing changes in social relationships can be integrated, like altruism, negotiation, mediation, delegation, and public rituals.

As forms of regulation based on social relationships, seeking and giving social support, as well as negotiation, have been studied. In particular, prosocial behavior or altruism that increases well-being [19]. Negotiation is aimed at making a deal and proposing a compromise with others, in order to reduce stress. Previous studies have identified negotiation and altruism as frequently chosen regulation strategies that are associated with adjustment [20]. 

Mediation and delegation are forms of regulation also linked to the use of social relationships. Mediation seeks the intervention of third parties as intermediaries, while delegation involves vicarious and indirect control through figures of higher status and expertise, in which the instrumental resolution of the situation is delegated (e.g., to a doctor, a teacher, or a more knowledgeable colleague). Mediation and delegation as indirect forms of changing the situation through social relationships were more effective than direct coping in situations with limited controllability [21]. However, the involvement of third parties as mediators and delegation were rarely used in previous samples, and the latter was not adaptive in the workplace [20].

Private and public rituals, as well as participation in social movements or social participation, constitute forms of regulation that involve the use of social instruments. Both private and public rituals have been associated with well-being in coping with collective stressful events [22]. In the work environment, it was identified the use of collective or public rituals as a regulation strategy [23] and there is evidence of their positive effects in other areas [18,24]. Finally, participation in social movements, although it is a less frequent form of coping, and is only applicable in situations of collective problems, can be an adaptive way for the regulation of stress [25].

## 3. The Use of Humor as an Effective Regulation Strategy

Another form of regulation that relates to social relationships is the use of humor. Styles of humour refer to how a person interacts with others using fun in interpersonal communications [26]. Humor mitigates negative affect, and together with the expression of affection o, constitutes a way of regulating feelings. Humor would aid in regulation through physiological reactions, producing relief through the production of endorphins, similarly to what occurs during physical activity [27]. Studies have shown that the activation of opposite emotions that influences the use of affection and humor directly helps to increase positive emotions and reduce negative ones [28]. Furthermore, humor can help to reinterpret and react to stressors in a more positive way, reducing their impact and symptoms of anxiety [27]. Research has also shown that humor reinforces adaptive regulation forms such as reevaluation or de-dramatization and the search for social support through affiliative humor [20]. Finally, there is a positive association between sense of humor, assertiveness, and creativity [28].

## 4. Emotional Regulation and Creativity

As mentioned earlier, positive forms of regulation such as humor are associated with creativity. In general, creativity can be defined as the ability to generate novel, original, infrequent, and useful or adaptive responses to a problem or situation. In the affective domain, creativity facilitates adaptive regulation. A personality trait linked to this topic is called emotional creativity (EC). It is the ability to attend to and experience complex emotions—rare or novel ones—in an authentic and adaptive way [29,30]. EC has been associated with well-being and correlated with a higher adjustment after an intense emotional experience. It has also been shown in university students to be a predictor of intrinsic motivation and academic commitment [31]. This trait is related to divergent thinking, the ability to be creative and original in emotional experiences, and emotional regulation, specifically reinforcing adaptive reappraisal. Finally, motional creativity is linked to creative performance [32].

In addition to this personality trait, forms of regulation such as humor, active physiological regulation, regulated emotional expression, gratitude and self-reward, thankfulness, and reappraisal facilitate creativity. The use of humor involves distancing, creative and positive reformulation of events. Probably through affective and cognitive pathways, it strengthens creativity or the generation of innovative responses [26]. Active physiological regulation, which involves some kind of physical activity, increases positive affect and, through it, creativity [33]. Regulated emotional expression increases creativity, probably because it is related to greater abilities to analyze and delay the response, which allows overcoming habitual responses [34]. Self-reward and thankfulness are associated with creativity because they involve the interpretation of reality in an innovative way, where the search for alternative positive stimuli helps to elaborate creative responses. Reappraisal as distancing and changing perspective also reinforces creativity [35].

## 5. Wellbeing and Emotional Regulation

Research has shown that people who use more functional regulation strategies experience high psychological well-being [4,36]. Therefore, well-being measures are also important to verify the functionality of emotional regulation strategies in different situations [37]. Specifically, some studies have focused on studying the relationship between regulation strategies and psychological or eudaimonic well-being (which refers to the extent to which individuals perceive their lives to be meaningful, purposeful, and fulfilling), while others have also related them to measures of hedonic well-being (which refers to the subjective experience of pleasure, happiness, and life satisfaction) [38].

For example, in a study conducted by [39] with 470 adults, positive reappraisal and refocusing on planning were positively correlated with subjective and psychological well-being. Additionally, rumination, catastrophizing, and self-blame were linked to poorer well-being. In another study low suppression, low self-critical use of humor, and affiliative humor were linked to higher levels of psychological well-being. Similarly, self-enhancing humor and low suppression were associated with higher levels of happiness [40].

More recently, a study conducted by [18] analyzed different strategies of emotional regulation [4] and its association with hedonic and eudaimonic well-being. The results showed that maladaptive coping strategies, such as withdrawal and social isolation, and suppression were associated with higher levels of negative affect or low hedonic well being. Conversely, coping strategies such as social support, gratitude and self-reward, reappraisal, as well as venting and regulated expression were found to be significantly associated with higher levels of positive affect. Similar results were found in the relationship of strategies with psychological well-being. Specifically, suppression, social isolation and withdrawal were associated with low psychological well-being., At the opposite, strategies such as problem-directed action and planning, attentional deployment through distraction, seeking social support, cognitive change by reappraisal, response modulation by active physiological regulation, as well as venting and regulated expression, acceptance and gratitude/self-reward, have been found to be positively correlated with psychological well-being. However, as observed in previous studies, the functionality of emotional regulation strategies and their relationship with well-being varies depending on the situations and interactions that occur with others [37]. Therefore, it is essential to conduct more studies that allow us to verify the functionality of emotional regulation strategies and their relationship with well-being in different contexts.

## 6. Present Study

Based on the reviewed literature, the objectives and hypotheses of this study are presented [31]. The shared objective was to compare how students, workers and athletes regulated emotions in the common domain of achievement or performance. In the presentation of the instruments, it was emphasized that we were asking how people managed emotions in the non-interpersonal and non-family environment, but in the area of performance or achievement, i.e., studies, work, and competitive sports. 

**Hypothesis 1 (H1).** *Differences in the frequency and effectiveness of strategy use are expected to be found between students, workers and athletes*.

The relationship between forms of affect regulation with adaptive goals of emotional regulation and well-being adjustment, health-related quality of life perception, humor, emotional creativity, and creativity or innovation in responses given to face a conflicting work situation will be contrasted. It is expected to find a congruent relationship between the use of functional strategies and the analyzed variables. 

**Hypothesis 2 (H2).** *Specifically, adaptive regulation is expected to be associated with stress adjustment, well-being and health related quality of life, as well as with trait creativity and innovation—the creativity trait is expected to be associated with creative performance in the form of innovative solution to a work conflict*. 

The specific effect of self and hetero-regulation strategies on adjustment will be examined, with well-being as a distal predictor and the strategies themselves as mediating variables. 

**Hypothesis 3 (H3).** *Adaptive regulation strategies will mediate between adjustment and well-being; with a direct relationship between both variables in students (see also [18])*.

## 7. Method

### 7.1. Procedure and Participants

Students and athletes data was collected by paper and pencil scales. For the study with employees, part of questionnaires were collected face-to-face and part online. Response time of the scales was 30 to 40 min. For data collected by encrypted mail link response rate was 30%. For face to face data response rate was close to 100% because they were controlled samples.

For the study with employees (E1), participants were recruited by doctoral students and professors of organizational psychology. Part of the questionnaires were collected face-to-face and part online. An encrypted email was sent to the participants who accessed a link to answer the survey. They could only reply once to the link that was sent to them—it was not a free access link. This study includes *N* = 687 individuals (workers, undergraduate and graduate students who work) from four countries: Brazil, Chile, Spain, and Uruguay, participated in this study. Of those who reported working (29%), 47% worked full-time and 53% worked part-time. Participants reported being professionals from different occupational fields (tourism, computer science, health care, teaching, administrative, civil servants, mechanics, HR and commerce technicians). 57% of the sample reported being female and 43% male, aged between 17 and 72 years (M = 30.77, SD = 10.87).

For the student’s study (E2), self-report paper and pencil scales were applied of line or face to face collectively, after requesting informed consent, in a single session by the researcher’s authors to young volunteers studying human sciences and education in different Spanish speaking institutions, who answered individually by self-applying the questionnaire. Participants were Spanish and Latin-American university students. The study included 959 students from Chile, Cuba, Spain, and Uruguay, who reported having completed high school or higher education [bachelor’s degree (64%), diploma (3%), degree (22%), and master’s degree (1%)]. 72% reported being female and 28% male, aged between 16 and 46 years (M = 20.67, SD = 2.87). In this study, the validation of the MARS scale was based on a first sample of students (see [18]).

For the study with athletes (E3), the self-report paper and pencil instruments were administered collectively, after requesting informed consent from the athletes who participated voluntarily. The participants were asked to answer the questionnaires thinking about the collective sport they most frequently practiced and in which they participated in competitions. *N* = 144 Spanish individuals who engaged in sports as a professional activity (athletics, cycling, football, rowing, rugby, surfing, basketball, others) participated in this study, 78% were men and 22% were women, aged between 17 and 43 years (M = 22, SD = 4.42). The validation of the MARS scale in sports was used in this study (see [24,37])

The inclusion criterion for all samples was to be of legal age and not to have cognitive limitations. The researchers who sent the links or collected the questionnaires checked the age and cognitive competence of each participant—who was personally known in each case. The three studies were developed in parallel during the 2015–2016 academic years as part of a research project on emotional regulation.

### 7.2. Instruments

The following figure and table presents the variables and scales applied in the studies (see Figure 1 and Table 1).

First, measures of self- (MARS and HSQ) and heteroregulation (EROS) were applied. Second, measures of stress adjustment and well-being (PHI). MARS and PHI were applied in all three studies. EROS only in students and athlete’s sample. Third, measures of health related quality of life (SF-36), trait creativity (ECI-S) and performance were only applied in study one. Fourth, styles of humor scales (HSQ) was applied only in study 2. The study design sought to examine the relationship of affective regulation to stress adaptation and well-being, as well as the relationship of affective regulation to trait and performance creativity.

In this research, instruments related to affective regulation were used. In this section, as a measure of affect regulation, the MARS scale [41,42] was responded to. A version consisting of 64 items was applied (see Table 2, Table 3, Table 4 and Table 5). Participants were asked to indicate how often they performed the actions mentioned to regulate/manage stress situations in the study, work, or sports context. The response scale was a Likert-type scale (0 = never to 6 = almost always). However, in the athlete’s studies, items of Altruism, Mediation, Negotiation, Delegation, Social participation, Search for information and Passive physiological regulation were not included—because they were not relevant for this domain.

Participants also answered the perception of adjustment in the mentioned episodes [17,40]. Here, the perceived functionality of the emotional regulation strategies was measured through six items that asked to what extent the following goals had been achieved when coping with stressful episodes at work: (1) decrease the intensity with which this emotional experience was lived, i.e., to change from high emotional experience, i.e., to change from high activation to calmness, relaxation calm, relaxation; (2) decrease displeasure, discomfort or unpleasantness, i.e., change from displeasure to greater pleasure; (3) understand the emotional experience of the situation; (4) to control or solve the problem associated with the situation, i.e., to change to greater control over the situation; (5) to manage relationships with other people, to change them for the better, to have a better relation (6) to improve one’s self-image. Responses were given on a Likert-type scale (1 = little or no change to 10 = change to a great extent). The reliability of the scale in the studies was highly satisfactory (α = 0.95 × 10^1^; α = 0.84 × 10^2^; α = 0.90 × 10^3^).

Other scales answered in this section by workers and student’s samples included the Emotion Regulation Questionnaire (ERQ) [43]. This scale assess individual differences in two emotion regulation strategies: cognitive reappraisal and expressive suppression. The scale consists of 10 items [e.g., when I want to feel more pleasant emotions, I change what I am thinking] and is responded to on a Likert scale (1 = completely disagree and 7 = completely agree). The reliabilities were found to be acceptable in the studies in which it was used (α = 0.74 and 0.69 × 10^1^; α = 0.77 and 0.78 × 10^2^). 

Participants in the students and athlete’s samples answer the emotional inter-regulation EROS instrument [44]. This scale of nine items evaluates individual differences in the regulation of others’ emotions (or hetero regulation). A first dimension composed of six items evaluates the tendency to increase others’ positive emotions and decrease their negative emotions [e.g., I did something pleasant or rewarding with others]. A second dimension of three items evaluates the tendency to increase others’ negative emotions [e.g., I talked to someone about their mistakes and limitations]. It is responded to on a Likert scale (1 = completely disagree and 5 = completely agree). The reliability in improving and worsening emotions was partially adequate in both studies (α = 0.76 and 0.61 × 10^2^; α = 0.75 and 0.89 × 10^3^).

Another section included in all studies consisted of measuring well-being and health-related quality of life. General, eudaimonic, hedonic, and social well-being were measured using the Pemberton Happiness Index (PHI) [45]. This instrument, which contains 11 items [e.g., “I feel very satisfied with my life”], is responded to on a Likert-type scale (0 = completely disagree and 10 = completely agree). The reliability was very adequate in all three samples (α = 0.87 × 10^1^; α = 0.87 × 10^2^; α = 0.91 × 10^3^). Health-related quality of life was measured using the SF-36 health survey in the study 1 [46]. This instrument is also responded to on a Likert-type scale. It includes a general health dimension that asks about physical health (1 = excellent to 5 = poor; from 1 = completely true to 5 = completely false), a mental health dimension that asks about anxiety and depression (1 = completely true to 6 = completely false), and a vitality dimension that asks about energy and fatigue (1 = always to 6 = never), in all cases during the last month. The reliability in this study [E2] was satisfactory (α = 0.75 for general health, 0.82 for mental health, and 0.82 for vitality).

In the study 2 humor styles were also measured using the Humor Styles Questionnaire (HSQ) [26]. This instrument consists of 32 items and four dimensions: affiliative humor (α = 0.40 if item 29 is eliminated), self-enhancing humor (α = 0.68), aggressive humor (α = 0.61 if item 23 is eliminated), and self-defeating humor (α = 0.70). Each dimension contains 8 items. It is responded to using a Likert-type scale (1 = strongly disagree to 7 = strongly agree).

The study 1 measured emotional creativity as a trait and its indicators of creativity and innovation in the ideation phase. Emotional creativity as a trait was measured using the Spanish version of the Emotional Creativity Inventory (ECI-S) [30], which is composed of 17 items grouped into three dimensions: emotional preparedness or disposition (e.g., when I have strong emotional reactions, I seek the reasons for my feelings); novelty or ability to experience new or unusual emotions (e.g., I have felt a combination of emotions that probably other people have never experienced), and effectiveness/authenticity (e.g., the way I express and experience my emotions helps me in my reactions with other people). Participants responded using a Likert scale (1 = strongly disagree, 6 = strongly agree). The reliability of this study was very satisfactory (α = 0.83, E1).

Participants in the study 1 were then presented with a case of a workplace conflict [32] and were asked to: (1) write about how they would feel in that situation, (2) how they would regulate their emotions and those of others in this case, (3) what they would include in a written proposal to the company’s management as solutions for improvement, and (4) what other actions they would take—independently of the written proposal—to resolve the workplace conflict. Using the CAT method [47], independent judges evaluated each participant’s fluency or number of ideas. They also evaluated the originality, effectiveness, and authenticity versus conventional or socially desirable responses using a Likert scale (5 = high, 1 = low). The correlations between the judges’ scores were significant with r_(63)_ = 0.65 for effectiveness, r = 0.70 for originality, and r = 0.68 for authenticity [E1].

### 7.3. Data Analysis

All the scales used have been translated and validated in Spain and Latin America, showing reliability and structural validity [32,48]. They are also formulated in standard international Spanish. In addition, cross-cultural value studies such as the World value survey show the existence of a shared Catholic European and Latin American cultural value domain [49]. This justifies in our opinion to carry out a general analysis. Reliability analyses were carried out for each scale per sample. Means comparison between samples of the use of regulatory strategies was carried out to contrast the exploratory H1. Analysis comparing how students, workers and athletes regulated emotions using only Spaniards samples shows similar results to general ones. Means of regulation strategies were compared between samples to explore differences between work, study and sport contexts. Subsequently, regulation strategies were correlated with stress adjustment and psychological well-being. Correlations between strategies with the variables of adjustment to stress and well-being in all samples allows us to examine H2. In more exploratory analysis, we examine the relationship of affect regulation with creativity in solving a work conflict. In the sample of workers, trait creativity and creativity of work conflict resolution were correlated with regulation strategies. In the student sample, the relationship between humor styles and regulation and criterion variables is examined. For a probability of a Type I error of α = 0.05 and a probability of a Type II error (1 − β) of 0.80, and for a typical effect size in social psychology of r = 0.21 [50] or of r = 0.24 [51] the required sample size for correlation analysis is *N* = 176 and *N* = 134 respectively [52] so that our samples meets the statistical power requirement. Finally, the mediational role of strategies between psychological well-being and stress adjustment is carried out to examine H3. A mediational analysis using Process was carried out in the student’s sample. For student large sample basic multiple regression analysis prerequisites (homoscedasticity...) were meet. Psychological well-being as global evaluation of person disposition is arguable the cause (X variable) of strategies of coping (M or mediation variables) as well a cause of adjustment to stress in the achievement domain (Y or effect or dependent variable). Moreover, PWB correlated with both coping strategies and adjustment, and coping with adjustment—meeting four steps that establish mediation X-M-Y are all correlated (see Kenny https://davidakenny.net/cm/mediate.htm, accessed on 23 October 2022) [53]. However, recall that Hayes states that these correlations need not be significant in order to apply a mediational analysis. In our case, we only included variables that were significantly correlated in any case. Program was Hayes’s Process. Effect size estimation for mediation was carried out as suggested by a reviewer with Kenny and another computer software. Using Kenny MedPower https://davidakenny.shinyapps.io/MedPower/, accessed on 23 October 2022, a sample of 250 is required for power 0.80 or higher in the direct and indirect coefficients of a mediational analysis (for an estimated effect size 0.20). For a probability of a Type I error of α = 0.05 and a probability of a Type II error (1—β) of 0.80, and for a mediation coefficient of r = 0.20, and supposing 10 covariables, a sample of *N* = 276 is required—using https://xuqin.shinyapps.io/CausalMediationPowerAnalysis/, accessed on 23 October 2022. So our sample meets the assumptions and statistical power requirement for mediational analysis.

## 8. Results

### 8.1. Frequency and Effectiveness of the Use of Regulation Strategies for Each Analyzed Context

The results showed the frequency and effectiveness of affect regulation strategies used in different domains. Figure 2 displays the affect strategy families that demonstrated reliability levels above 0.65 in at least one of the three samples.

The strategies that have been found in previous studies to be dysfunctional or non-adaptive (e.g., abandonment and suppression) show lower means than adaptive ones (Figure 1 and Table 2, Table 3, Table 4 and Table 5), demonstrating that people tend to report a functional profile of emotion regulation.

### 8.2. Modification of the Situation and Social Relationships

This facet includes approach versus avoidance of social situations and relationships (Table 2 and Table 3). Analysis of variance examines exploratory H1. Athletes (E3) followed by students (E2) use more instrumental coping and seeking instrumental and informational support than workers (E1). Effect sizes were small, differences between groups explaining between 1 and 2% of variance. In turn, student’s use more emotional support seeking strategies than workers. In addition, students followed by worker’s report using talking to someone who could give me advice and guidance more than athletes. The family of behavioural disengagement and social isolation is used little as a strategy, differences between groups are very small (Table 2). Regarding the other families or strategies of social bonds (Table 3) there are no major differences in altruism and mediation. However, students use more negotiation. The greatest difference between workers, students, and athletes appears to be in the use of private and public rituals—group differences explain 27 and 6% of variance.

### 8.3. Attentional Deployment and Cognitive Change

By respect to Attentional Deployment and cognitive change (Table 4) athletes use this facet more intensely than the other two groups. Specifically, the sample of students alternating with athletes in some items versus workers are the ones who use distraction the most- 12% and 19% of variance was explained. For example, while students seek to do something entertaining, something that I really like and enjoy, athletes use thinking about something else to distract myself from my feelings. Athletes, followed by workers, show more use of the gratitude and self-reward strategy than students, except in doing something special to reward and feel better, this strategy is more used by the last group. Explained variance was 8.6%, 5% and 2,4% for items 28, 30 and 31 respectively. Athletes, followed by workers, more than students, report using spiritual activities and reappraisal as a form of regulation—but only in one item and explaining 1.8% of variance in the last case. Athletes report performing more social comparison and students less, with workers located between both groups in the use of this family of regulation—explained variance was 2.5% and 9.2%. 

### 8.4. Experience Modification and Regulation of Emotional Response

Regarding experience modification and regulation of emotional response (see Table 5) workers report using the inhibition/suppression dimension mostly, followed by students and workers using discharge- explained variance was low, around 1%. Compared to the other two groups, athletes use this regulation strategy less. Regarding active physiological regulation, it is athletes who report using this family of regulation the most, and students who use it the least. Worker scores are in an intermediate position. Regarding passive physiological regulation, workers mention using it more frequently than students—explained variance was important around 5%. Finally, athletes—followed by students—report greater use of confrontation as a regulation strategy than workers—explained variance was of 6.7% and 1.2%.

### 8.5. Relationship between Adaptive Emotional Regulation, Well-Being, and Stress Adjustment

Correlation was carried out between strategies and adjustment to stress and wellbeing to examine H2. Positive associations were found for functional strategies and negative associations for dysfunctional strategies (italicized) as expected (Table 6). Of the forms of social regulation that are based on mobilizing social relationship, in addition to seeking social support, negotiation is shown to be adaptive in this study, as it is linked to both adjustment and well-being in workers and students, and to a lesser extent, altruism, which is associated with adjustment in both groups but not well-being. Participation is associated with adjustment and well-being in students, while in workers it is negatively associated with both. Rituals are adaptive in athletes but not in students and are negatively associated with well-being in workers. Finally, mediation and delegation do not confirm their supposed adaptive character, and the latter is even negatively associated with well-being in samples of workers and students.

The distraction strategy is positively associated with both adjustment and well-being in students and athletes; acceptance is positively associated with adjustment in athletes and with well-being in both groups. These are attentional reorientation and cognitive change strategies, that are partially adaptive in this study. Rumination and social comparison are non-adaptive strategies in students and workers, although they are positively associated with adjustment and well-being in athletes (see Table 7).

Active physiological regulation shows its adaptive character and passive regulation shows its dysfunctionality as expected. The use of humor and demonstration of affection to regulate emotional response is adaptive in workers and students. In turn, regulated emotional expression is adaptive with adjustment in all three groups and with well-being in students and athletes. Confrontation partially confirms its negative association with well-being, although this does not occur with discharge, which, contrary to expectations, is associated with well-being in students. The results obtained confirm part of the second hypothesis proposed (see Table 8).

Finally, heteroregulation by improvement of emotions of others measured by EROS-enhancement was related to adjustment and wellbeing, but only in one sample: students: r = 0.21, *p* = 0.01, and athlete’s r = 0.03 n/s for adjustment; and students: r = 0.24, *p* = 0.01, and athlete’s r = −0.07 n/s for well-being. Surprisingly, heteroregulation by increasing others’ negative emotions was also related to adjustment and wellbeing in the students sample—EROSnegative correlated in students r = 0.10, *p* = 0.05, and athlete’s r = 0.04 n/s for adjustment; and students: r = 0.11, *p* = 0.05, and athlete’s r = 0.12 n/s for well-being.

### 8.6. The Relationship between Adaptive Regulation Strategies, Humor Styles, Creativity, and Innovation

Regarding humor styles, self-enhancing humor, showed stronger functionality as expected, being associated with well-being—and reappraisal, although also with discharge. The aggressive humor dimension, which is assumed dysfunctional, is effectively associated with poorly adaptive strategies such as confrontation, delegation, passive physiological regulation, and discharge. However, questioning the idea that aggressive humor is dysfunctional, in this study, it is associated with adaptive strategies such as seeking support, distraction, self-reward, reappraisal, and altruism. Self-defeating humor was linked to non-adaptive regulation strategies such as suppression, passive physiological regulation, delegation, and confrontation. However, despite being non-adaptive, this humor style was associated with reappraisal, active physiological regulation, and altruism, which are positive regulation strategies (see Table 9).

Supporting the scale’s validity, there is an association between emotional creativity (ECI-S scale) and creativity in response to a work conflict (E1). In turn, emotional creativity as a trait is positively and significantly associated with adjustment and strategies linked to high well-being (direct coping, social support, re-evaluation, spirituality, information seeking, active physiological regulation, regulated emotional expression), as well as with acceptance, social participation, and rituals. It is also positively associated with non-adaptive strategies such as rumination, social comparison, and discharge, and negatively associated with suppression. In this study, creativity, or innovation in the ideation phase as a response to a work conflict is positively and significantly associated with re-evaluation, active physiological regulation, and information seeking. It is negatively associated with suppression. Finally, functional forms of regulation (direct coping, negotiation, reevaluation, active physiological regulation) are positively and dysfunctional (abandon, rumination, passive physiological, suppression) are negatively v associated to health-related quality of life in workers (SF-36 scale). 

### 8.7. Effect of Self-Regulation Strategies as Mediator Variables with Well-Being and Adjustment

To examine H3 mediational analyses [54] were performed with the sample of students, using well-being as predictor, adjustment as the dependent variable (DV), and regulation strategies by area as mediator variables (MV). Well-being and regulation strategies explain 28% of the variance in adjustment. Well-being is positively and indirectly associated with adjustment through direct coping *B* = 0.023, SD = 0.0052, 95% CI [0.014; 0.035] and lower psychological abandonment *B* = 0.023, SD = 0.0073, 95% CI [0.009; 0.038]. This association is also observed through positive reevaluation *B* = 0.017, SD = 0.004, 95% CI [0.093; 0.028], active physiological regulation *B* = 0.002, SD = 0.0017, 95% CI [0.0003; 0.0073], and regulated expression *B* = −0.0044, SD = 0.0023, 95% CI [−0.010; −0.001]. On the other hand, well-being is negatively associated with adjustment through acceptance *B* = −0.0044, SD = 0.0023, 95% CI [−0.010; −0.001] and discharge *B* = −0.0048, SD = 0.0022, 95% CI [−0.010; −0.0014]. The indirect effect is 0.0656 and explains 42% of the total effect (see Figure 3).

## 9. Discussion

The three hypotheses postulated in this study were partially confirmed. Differences in the use and, to a lesser extent, the efficacy of regulation strategies were found between workers, students, and athletes. Dysfunctional regulation strategies such as rumination and social comparison are maladaptive for workers and students but not for athletes, which has also been shown in previous studies [18]. Moreover, maladaptive strategies in the former two groups such as confrontation and suppression do not have negative effects on athletes (H1). 

There is a congruent relationship between the use of functional strategies and adjustment to stress, well-being, and health-related quality of life. This congruence is also found to some extent in humor styles such as self-affirmation, as well as in creativity, both emotional and applied or innovation in the ideation phase to cope with a conflicting work situation (H2). However, negative heteroregulation, that worsens the emotions of others, is associated with well-being and personal adjustment in the student sample. This result may suggest that attempting to influence the emotions of others, even critically, has positive effects for the individual, reinforces self-esteem and personal self-efficacy, even if it has negative effects on others. However, this conclusion must be relativized because of the low reliability of the scale. 

Furthermore, it was confirmed that adaptive regulation strategies mediate the relationship between well-being and adjustment to stress (H3). It is possible to conclude that the use of adaptive self- and hetero-regulation strategies facilitates adjustment and is related psychological well-being. Confirming their general adaptive nature, the following strategies were associated with adjustment and well-being: direct coping, social support, distraction, self-reward, gratitude, reappraisal, active physiological regulation, use of humor and affection, as well as regulated expression. On the other hand, low psychological abandonment, low passive physiological regulation, and low suppression were associated with adjustment and well-being. The results suggest that individuals who know and regulate their own emotions properly can be more effective in work, studies, and sports teams. This is consistent with a recent meta-analysis showing the relevance of these emotional processes in adaptive behavior in different contexts [55]. A state of high well-being has horizontal influence between peers, and vertical influence through the leaders, helping to create a positive group climate. Since moods tend to be contagious and/or transmitted in both work teams and groups, when they are positive in a high well-being climate, they can help mitigate negative situations in study, work, and sports groups [56]. When leaders are in a good mood, team/group members are more positive and cooperate more [57]. These aspects can facilitate their creativity [48,58].

In the following paragraphs we discuss, in detail, the difference in frequency of use of the strategies between students, workers and athletes. We then examine their functionality in each group. We differentiate in our discussion the three phases of Gross’s model: situation modification and social bonding, attention reorientation and cognitive change, and finally, experience modification and emotional regulation.

### 9.1. Differences between Groups on the Frequency of Strategies: Modification of Situation and Social Relationships

Regarding the facet of modification of the situation and social relationships, it was found that *athletes use problem-solving and social support-seeking strategies (especially instrumental and informative) more than workers*. The effect size was small, of eta squared 0.01 or 0.02, equivalent to an r = 0.10 or 0.14 or *d* = 0.20 or 0.28. These results could be explained by the fact that direct action (see items 04, 05 and 06 in Table 2) is possibly more frequent in the professional sports context, although it is also expected to be commonly used in the educational context and desirable in the workplace. Lower use of social support strategy in the worker sample than in the student sample, is probably explained by the fact that the work environment tends to be more concrete and focused on the individual than that of sports—especially if it is cooperative or collective.

*Students specifically seek more emotional social support*, and express and discharge emotions more intensely than the other two groups, as described below. This could be explained by a more informal *youth sociability and more horizontal social relationships.* It is possible that in work and high-performance sports contexts, more regulated interactions give less room for expressive spontaneity. 

Overall, *students use more regulation forms that involve mobilizing social relationships* in line with more horizontal and spontaneous interactions that occur in the educational setting. Specifically, students make relatively more use—than workers- of negotiation. In turn, *athletes make greater use of rituals as a regulation strategy than the other two groups.*

### 9.2. Modification of Situation and Social Relationships: Searching Social Support, Altruism and Negotiation as Adaptive Strategies and Mediation and Delegation as Inadpative Strategies

The results show that *direct coping, seeking social support, and low avoidance are adaptive emotional regulation strategies*. In addition, some forms of social regulation, like helping and negotiating with others are effective forms of regulation. *Altruism and negotiation are associated with adjustment and well-being in students and workers*. This result confirms that helping others allows individuals to move away from egocentrism and feel effective and socially integrated [20]. Helping others, even if done “selfishly”, is effective [19]. Negotiation may be adaptive because it involves negotiating at a similar level of status and reducing conflict.

Others forms of social regulation, *like mediation and delegation are negatively associated with adjustment in workers, but not in students*. The results suggest that asking for a third-party intervention in the work context is not functional, probably because it implies “breaking” defined roles. Delegating to others is also not adaptive in either group. In performance or achievement contexts, delegating is associated with not directly taking control of the situation, which can result in low levels of effectiveness and self-esteem. 

### 9.3. Social Participation, Private and Public Rituals and Adjustment; Adaptives in Students and Athletes Respectively

Social participation shows a positive association with adjustment and well-being in students, while in workers, it is negatively associated with well-being. These results suggest that despite the current weaknesses of the student movement, participating in social movements has positive effects for students. As for workers, the low well-being in relation to social participation could be explained by the weakness of the labor movement and the cost it could have for salaried personnel.

While *rituals are not adaptive in students and show a negative association in workers, they are associated with well-being and adjustment in athletes*. Evidence supports that the level of collective ritual and its effects are potent in sports contexts where the use of amulets, lucky signs, coordinated chants, and collective celebrations are behaviors of high frequency and significance [59]. It has been postulated that the use of private rituals is common for coping with competition anxiety [60,61]. This study confirms this approach, suggesting that using rituals as a regulation strategy reduces anxiety and helps to focus on the task, allowing for an improvement in performance.

### 9.4. Differences between Groups on the Frequency of Strategies: Attentional Deployment and Cognitive Change

Regarding the aspect of attention deployment and cognitive change, it was found that *students and athletes report using distraction more often than workers* as a strategy. *Athletes use self-reward, gratitude, reevaluation, and social comparison more frequently* than the other two groups.

*Workers and athletes use spiritual activities strategies more than students.* This shows a greater secularization in young people with higher education than in other groups.

Athletes reevaluate and *self-reward more and have a higher attitude of gratitude than the other two groups*—and the effect size was medium high or important. The fact that this group uses all forms of cognitive change and focus on activity corroborates what has been found in other studies [62,63]. Although sport is found to be a source of greater satisfaction and flow than study and work [64], other results qualify this assertion [65]. It is possible that in the study environment and some work contexts, it is more difficult to find and use positive stimuli to reward oneself. Another explanation is that people may not know how to use these regulation styles in the analyzed areas, even though they are familiar with them.

### 9.5. Attentional Deployment and Cognitive Change: Reevaluation, Gratitude and Self-Reward as Adaptive Strategies and Rumination and Social Comparison as Inadpative Strategies in Workers and Students

*Reevaluation, gratitude, and self-reward were associated with adjustment and overall well-being* in general, while *distraction* was only associated with these variables in *students and athletes*. *Distraction is not adaptive in the workplace* probably because the cost of redirecting attention by ignoring problems instead of focusing on them to solve them is probably higher than in the other two areas.

*Rumination and social comparison are non-adaptive strategies in students and workers, although they are positively associated with adjustment and well-being in athletes* in this and other study [66]. Although workers report using social comparison at a moderate level, it is negatively associated with well-being in this group. This last result is also found in students. Remember that in general, social comparison has been shown to be a maladaptive regulation strategy [18]. However, in a competitive context such as team sports, ruminating on thoughts and comparing oneself to others seems to motivate one to perform and feel better as a result.

### 9.6. Differences between Groups on the Frequency of Strategies: Experience Modification and regulation of Emotional Responses

In the facet of modification of experience and emotional response, it has been found that *workers use more inhibition and suppression of emotions than students and athletes*, probably due to the greater demands for emotional self-control in the workplace. However, it is important to remark that effect size was small.

*Athletes are the ones who report using active physiological regulation more* than the other two groups, in line with the fact that the activity they engage in is predominantly physical—effect size was strong and explained variance was 5% or more. Workers use this form of regulation at an intermediate level, and also report using passive physiological regulation more than students—effect size was small. This last result could probably be explained by age and suggests that workers make a greater effort at regulation, although it may not be more adaptive (in the case of passive regulation). 

*Students and workers report engaging in more discharge, but athletes confront more than students and workers-* effect size were from small to medium high. This probably reflects the relative greater acceptance of expressiveness in the study/work environment and the greater need for assertiveness and competitiveness in the sports field.

### 9.7. Experience Modification and Regulation of Emotional Response: Active Physiological Reaction, Regulated Emotional Expression as Adaptive Strategies and Passive Physiological regulation and Suppression as Inadaptive Strategy

Active physiological regulation shows its adaptive character and passive regulation shows its dysfunctionality as expected. Regulated emotional expression is adaptive in all three groups. Previous studies in this field have shown that self-regulation and regulation of others through suppression are associated with low emotional intelligence and poor psychological adjustment [67]. These results are replicated in this study, except for the case of athletes. While these strategies have been shown to be maladaptive in other studies, it was found that *both discharge and confrontation were associated with regulated expression of emotions and adjustment in the sample of workers*. These results suggest that direct but non-aggressive expression of anger may be an adaptive strategy in the work environment [32]. On the other hand, the fact that venting is not dysfunctional in sports can be explained, in part, because an ideal emotional profile in sports included, among other traits, being somewhat anxious and a little angry, consistent with the meta-analysis of [68] on sport psychology and performance.

### 9.8. Modification of Response of Emotions by Humor and Affection as Adaptive Strategy in Workers and Students

Results show that the *use of humor and affection is associated with adjustment and well-being in workers and students, although not in athletes*. This could suggest that the expression of humor and affection plays different roles depending on the analyzed context. It is possible that the sense of competition and commitment in sports leads to humor being considered a distraction -as measured in this study- and that demonstrations of affection are not desirable or are expressed differently in the context of competitive sports, studying, and work. It is important to remember, in the same vein, that there was less acceptance of expressiveness or discharge and greater confrontation in the sports field than among students and workers. Humor styles were also found to be associated with well-being. Consistent with previous studies [69], in this study, *self-enhancing humor was associated with well-being*. Aggressive humor was associated with non-adaptive coping strategies such as confrontation and discharge, suggesting a social integration cost for this style. However, this humor style was also associated with altruism, gratitude, and distraction, which help to obtain social support. Its ambivalent nature can be explained by the fact that it is not negatively associated with well-being. Self-defeating humor, which is usually associated with distress neuroticism and low wellbeing [26], was found to be associated with non-adaptive strategies such as suppression, passive physiological regulation, confrontation, and delegation, which may explain its general association with distress. However, it was also associated with reappraisal and active physiological regulation. The ambivalent nature of self-defeating humor can be explained by the fact that this style focuses on internal processes. Laughing at oneself as a detached critique can be adaptive, while coping with stress through intense physical exercise can be an externalized form of coping. However, the limitations of the sample size relativize these conclusions.

### 9.9. Adaptive Regulation like Reevaluation and Active Physiological Regulation Are Related to Emotional Creativity and Creative Solution of Work Conflict

Regarding the relationship between regulation and creativity, it was found that *reappraisal and active physiological regulation are associated with emotional creativity as a trait and applied to a task.* Emotional creativity trait is associated with innovative responses to solve a labor conflict, confirming that it predicts creative performance. The fact that emotional creativity is associated with regulation strategies that also relate to well-being, and that both indicators correlate with adjustment, suggests that creativity is adaptive and helps people to successfully coping with stress and managing emotions. The results indicate that, in a situation of labor conflict, people who give more creative responses did not choose dysfunctional regulation strategies such as abandon or giving up, or acting as if nothing happened. They also do not suppress their emotions and report more positive reappraisal—which helps personal growth -, use greater active physiological regulation, and information seeking. The results corroborate that *low abandonment, finding the positive side of the situation, physical activity, high trait emotional creativity*, *and low suppression* make up an optimal functioning profile for creativity.

### 9.10. Wellbeing as Dispositional Variable Predicting Adjustment through Adaptive Regulation—Direct Coping, Low Abandonment, Reappraisal, Active Physiological Regulation, and Regulated Expression

The analyses confirmed that well-being predicts an adaptive coping profile, and that these partially mediate between well-being and adjustment, explaining between 10 and 20% of variance. The specific strategies that mediate between these variables are direct coping, low abandonment, reappraisal, active physiological regulation, and regulated expression. These confirm their central role in emotional regulation. Well-being was associated with greater acceptance and discharge, which in turn showed a negative effect on adjustment in this analysis. That is, these strategies mediated negatively. The results suggest that students with higher well-being tend simultaneously to express their emotions intensely while also practicing self-control. Multivariate results suggest that, when eliminating the influence of other adaptive regulation forms, these strategies (discharge and acceptance) have a negative influence, e.g., if the influence of regulated expression of emotions is eliminated, discharge itself does not help adjustment. Acceptance, if the influence of positive reappraisal is eliminated, is not helpful either. These results were found with nuances in the other two groups (mediational analysis not shown). For example, in workers, the effect of well-being on adjustment was shown to be significant through gratitude, low discharge, and high reappraisal. In athletes, acceptance mediated positively between well-being and adjustment. 

As study limitations, it is noted that this is a cross-sectional study that prevents making causal statements and that the samples were large but convenience samples. Another limitation is that the reliabilities of some of the scales, like EROs worsening heteroregulation scale and affiliative humor scales—in some of the samples in this study—are low. MARS strategies such as behavioral avoidance and distraction in athletes, rumination and active physiological regulation in students, showed reliabilities below 0.70, although not as low as the previous ones. This limits the scope of the results of this study. Regarding the results of humor and creativity, the small samples do not allow for generalizing the findings, however, they are consistent with previous studies. Future studies need to consider these limitations when comparing different nations and exploring whether regulation strategies are influenced by cultural patterns.

## 10. Conclusions

In conclusion, it is noted that in general, an adaptive regulation profile would be formed by high direct coping, which would help people perceive and validate effective actions. This profile would also involve focusing attention on the positive, self-rewarding, gratitude attitudes, distracting from the most negative and stressful aspects, and focusing attention on rewarding stimuli. Not inhibiting emotions and expressing them in a regulated manner is another component of the adaptive profile, as well as thinking positively or reevaluating the situation by distancing oneself from stress and extracting positive aspects from what has happened. Regarding hetero-regulation, in general, adaptive tactics are direct coping, reappraisal, and social support for managing the emotions of others. People like their leaders to motivate them and give positive feedback for a job well done. The mentioned profile of self and hetero-regulation can contribute to high well-being. Specifically, this profile should be examined in different samples and their specificities, e.g., criticism and expression of dissatisfaction with the emotions of others in the sports context are associated with well-being, suggesting that in that context, they are adaptive. Training and intervention in the area of affective regulation should take into account the predominance of functional emotional social influence on others, which contributes to generating emotionally positive climates with low negativity.

## Figures and Tables

**Figure 1 ijerph-20-06865-f001:**
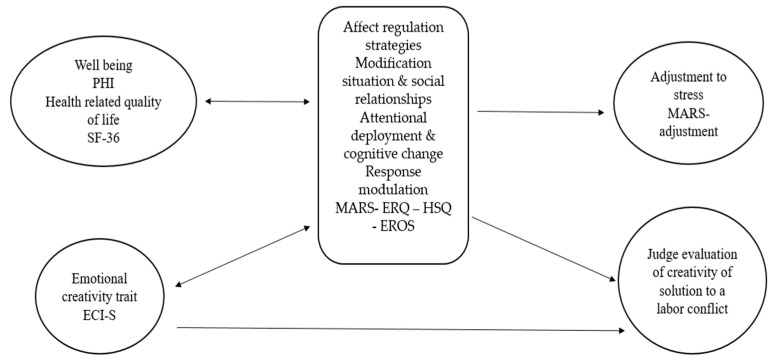
Study model: dispositional variables (psychological well-being and health-related quality of life and trait emotional creativity), mediating variables (regulation) and criterion variables (adjustment to stress and applied creativity or innovation as a solution to a labor conflict).

**Figure 2 ijerph-20-06865-f002:**
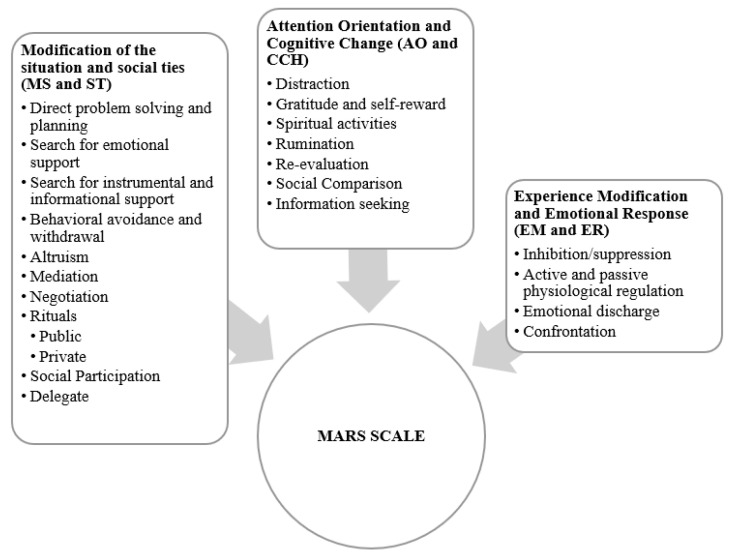
Facets and regulation families of Measure of Affect Regulation Styles, (MARS).

**Figure 3 ijerph-20-06865-f003:**
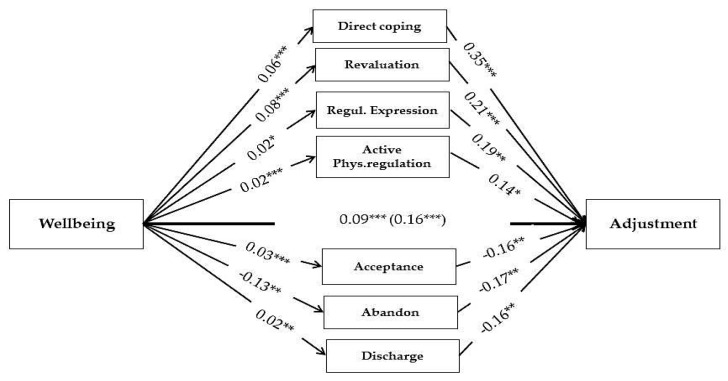
Relationship between well-being (measured with PHI) and stress adjustment in a sample of students (E2). Emotional regulation strategies are the mediators (MARS scale). The coefficients are not standardized. The total (and direct) effect of well-being on adjustment is shown. * *p* < 0.05; ** *p* < 0.01; *** *p* < 0.001.

**Table 1 ijerph-20-06865-t001:** Variables, scales and studies included in the study.

Variable	Scale	Applied in Study
**Affect regulation scales**		
Affect self regulation	MARS strategies	1, 2 and 3 (7 items not included in E3)
	ERQ reappraisal and suppression	1 and 2
	HSQ Humor styles	2
Affect Heteroregulation	EROS	2 and 3
Criterion and covariates variables		
Ajustment to stress	MARS-adjustment	1, 2 and 3
**Wellbeing and health**		
Psychological well-being	PHI	1, 2 and 3
Health related quality of life	SF-36	1
**Creativity**		
Trait emotional creativity	ECI-S	1
Emotional creativity performance	Solution to a labor conflict evaluated by judges in creativity	1

**Table 2 ijerph-20-06865-t002:** Reliabilities (α) of the dimensions, mean, standard deviation (SD) and one-way analysis of variance of situation modification and social ties items.

Measure	Study 1 Workers	Study 2 Students	Study 3 Athletes	*F* (2, 1787)	*p*	η²
**Direct problem solving and planning. Instrumental coping**	α = 0.76	α = 0.73	α =0.71			
04. Develop a plan to deal with what happened and be able to do something to change the situation	3.80 (1.54)	4.04 (1.45)	4.18 (1.53)	6.9	0.001	0.007
05. Act or do something to solve the problem that caused my mood	4.10 (1.39)	4.24 (1.25)	4.78 (1.39)	15.94	0.0001	**0.02**
				(1, 1644)		
06. Make plans or make a decision to avoid similary problems in the future	4.14 (1.32)	4.32 (1.28)	n.d.	7.71	0.0056	0.005
**Seeking emotional support**	α = 0.92	α = 0.80	n.d.			
52. Talking to someone about how I feel.	3.84 (1.70)	4.10 (1.64)	n.d.	9.76	0.018	0.006
53. Speak up for understanding and support	3.52 (1.78)	3.91 (1.71)	n.d.	19.99	0.0001	**0.012**
**Search for instrumental and information support**	α = 0.88			(2, 1787)		
54. Asking someone else for help in resolving the situation that triggered my mood	3.10 (1.72)	3.31 (1.78)	3.40 (1.77)	3.54	0.03	0.004
55. Talking to someone who could give me advice and guidance	3.61 (1.66)	3.84 (1.67)	3.50 (1.92)	4.88	0.007	0.005
56. Ask someone who faced a similar problem what they did	3.23 (1.71)	3.34 (1.17)	4.01 (1.81)	17.16	0.0001	**0.019**
**Behavioral avoidance and withdrawal**	α = 0.72	α = 0.69	α = 0.67			
07. Leaving or leaving the situation	1.63 (1.46)	1.70 (1.38)	1.89 (1.60)	2.04	0.13	0.002
08. Acting as if nothing is wrong	1.62 (1.42)	1.88 (1.53)	1.71 (1.53)	6.22	0.002	0.007
09. Giving up or doing nothing: not trying to control the situation	1.15 (1.25)	1.21 (1.29)	1.15 (1.21)	0.50	0.60	0.0001
				(1, 1644)		
13. Avoid contact with people associated with the problem	2.74 (1.72)	2.71 (1.70)	n.d.	3.96	0.046	0.002
14. Seeking to be alone	2.30 (1.76)	2.63 (1.70)	n.d.	14.64	0.0001	0.009
35. Try to accept my fate, which is inevitable, it has to be this way	2.33 (1.86)	2.70 (1.80)	n.d.	16.45	0.0001	0.001

Note. n.d. = no data available in this study. In bold the eta squares that explain 1 percent or more of the variance.

**Table 3 ijerph-20-06865-t003:** Reliabilities (α) of the dimensions, mean, standard deviation (SD) and one-way analysis of variance of strategies measured with a single item in the Modification of Situation and social relationships facet.

Measure	Study 1 Workers	Study 2 Students	Study 3 Athletes	*F* (1, 1644)	*p*	η^2^
**Altruism**						
41. Forget about my situation and help someone else	2.37 (1.53)	2.45 (1.49)	n.d.	1.13	0.28	0.001
**Mediation**						
57. Ask someone to mediate or intervene in relation to what happened	1.55 (1.50)	1.64 (1.54)	n.d.	1.37	0.24	0.001
**Negotiation**						
58. Talk about what happened with the people involved to negotiate or reach an agreement	2.92 (1.68)	3.28 (1.63)	n.d.	19.03	0.0001	**0.011**
**Rituals**				(2, 1787)		
**Private**						
59. Participate in or organize a private ceremony (I wrote an e-mail about what happened, reorganized photos, letters)	1.00 (1.42)	0.91 (1.41)	4.09 (1.48)	325.24	0.0001	**0.27**
**Public**						
60. Participate on or organize a public ceremony (demonstration, mass, commemoration)	0.62 (1.25)	0.65 (1.28)	1.89 (2.08)	56.99	0.0001	**0.06**
**Social Participation**				(1, 1644)		
62. Get involved in political or social activities	1.24 (1.53)	1 (1.55)	n.d.	9.7	0.002	0.006
**Delegation**						
64. Putting myself in the hands of others to solve my problem or improve the situation	1.12 (1.32)	1.32 (1.44)	n.d.	8.27	0.004	0.005

Note. n.d. = no data available in this study. In bold the eta squares that explain 1 percent or more of the variance.

**Table 4 ijerph-20-06865-t004:** Reliabilities (α) of the dimensions, mean, standard deviation (SD) and one-way analysis of variance of attention orientation and cognitive change items.

Measure	Study 1 Workers	Study 2 Students	Study 3 Athletes	*F* (2, 1787)	*p*	η^2^
**Distraction**	α = 0.81	α = 0.78	α = 0.62			
21. Doing something fun, something I really like and enjoy	4.07 (1.48)	4.28 (1.40)	3.66 (2.30)	11.92	0.0001	**0.013**
				(1, 1644)		
22. Watch TV, read a book, listen to music, etc., to distract myself	4.07 (1.55)	4.35 (1.43)	n.d.	14.3	0.0002	0.009
				(2.1787)		
23. Working on something to keep busy to take my mind off my mood	3.73 (1.52)	3.75 (1.54)	3.82 (2.21)	0.19	0.83	0.0001
24. Thinking about something else to distract me from my feelings	3.47 (1.48)	2.37 (1.48)	4.73 (1.51)	208.85	0.0001	**0.19**
				(1, 1644)		
25. Being with people, talking, to forget my mood	3.60 (1.57)	3.83 (1.56)	n.d.	8.85	0.003	0.005
**Gratitude and self-reward**	α = 0.68	α = 0.71	α = 0.82	(2, 1787)		
28. Do something special to reward myself and feel better	2.97 (1.71)	3.03 (1.68)	4.89 (1.39)	84.58	0.0001	**0.086**
30. Trying to think about those things that I’m doing well at	3.60 (1.60)	3.34 (1.51)	4.69 (1.51)	48.41	0.0001	**0.05**
31. Trying to be grateful for the things that are going well in life	3.95 (1.63)	3.84 (1.68)	4.83 (1.88)	21.82	0.0001	**0.024**
**Spiritual activity**	α = 0.88	α = 0.82	n.d.			
36. To pray, to put my faith in God, to lean on religious things	2.20 (2.19)	1.59 (1.97)	3.32 (2.09)	51.18	0.0001	**0.054**
				(1, 1644)		
40. Read or do something religious, spiritual	1.87 (2.03)	1.07 (1.64)	n.d.	77.94	0.0001	**0.045**
**Rumination**	α = 0.64	α = 0.57	α = 0.76	(2, 1787)		
01.Think about how you could have done things differently	3.88 (1.41)	4.09 (1.29)	4.28 (1.51)	7.58	0.0005	0.008
02. Trying to understand my feelings by thinking about and analyzing them	4.04 (1.37)	3.97 (1.48)	4.17 (1.57)	1.39	0.24	0.0015
03. Repeatedly thinking about what happened, about the emotional effects of the situation	3.77 (1.46)	3.95 (1.48)	4.00 (1.60)	3.42	0.03	0.004
**Re-evaluation**	α = 0.81	α = 0.80	α = 0.77			
37. Try to reinterpret the situation, to find a different meaning or sense	3.40 (1.54)	3.31 (1.54)	3.91 (1.67)	9.38	0.0001	0.01
38. Trying to see things from a broader perspective	4.04 (1.37)	3.78 (1.35)	4.41 (1.51)	16.77	0.0001	**0.018**
39. Trying to find somethings good in the situation	3.87 (1.49)	3.76 (1.51)	4.00 (1.60)	2.16	0.11	0.002
**Social Comparison**	α = 0.69	α = 0.60	α = 0.59			
42.Comparing myself to people who are worse off than I am.	3.00 (2.78)	2.42 (1.71)	3.45 (1.91)	22.47	0.0001	**0.025**
43. Comparing myself to a person with more a means, personal resources and who had done better than me. To take him/her as a model to improve my situation	2.39 (2.58)	1.94 (1.62)	4.40 (1.87)	90.13	0.0001	**0.092**
**Information search**	n.d.	n.d.	n.d.	(1, 1644)		
61. Inform me about my problem or about the situation in order to overcome it or do better	2.93 (1.73)	2.78 (1.75)	n.d.	2.97	0.085	0.002

Note. n.d. = no data available in this study. In bold the eta squares that explain 1 percent or more of the variance.

**Table 5 ijerph-20-06865-t005:** Reliabilities (α) of the dimensions, mean, standard deviation (SD) and one-way analysis of variance of attention orientation and cognitive change items.

Measure	Study 1 Workers	Study 2 Students	Study 3 Athletes	*F* (2, 1787)	*p*	η²
**Inhibition/suppression**	α = 0.77	α = 0.66	α = 0.64			
10. Try not to think about what happened, ignore negative emotions	2.44 (1.68)	2.16 (1.44)	2.03 (1.54)	8.27	0.003	0.009
11. Try not to show my feelings, to suppress any expression of them	2.64 (1.72)	2.32 (1.68)	2.15 (1.59)	9.33	0.0001	**0.01**
12. Faking or expressing emotion opposite to what you feel	2.24 (1.73)	1.89 (1.60)	1.81 (1.50)	10.33	0.0001	**0.011**
**Physiological regulation**						
**Active**	α = 0.69	α = 0.45	α = 0.63			
15. Exercise, sport	3.02 (1.95)	2.62 (1.95)	4.40 (1.35)	56.06	0.0001	**0.059**
16. Practicing relaxation, meditation	2.61 (3.05)	1.80 (1.71)	3.58 (1.45)	50.68	0.0001	**0.054**
**Passive**	α = 0.79	α = 0.63	n.d.	(1, 1644)		
17. Sleeping or napping	3.35 (3.15)	2.76 (1.83)	n.d.	22.87	0.0001	**0.014**
18. Eating something to overcome my mood	3.24 (3.19)	2.61 (1.82)	n.d.	25.7	0.0001	**0.015**
19. Drinking coffee, caffeinated beverages or tea	2.36 (3.08)	1.77 (1.84)	n.d.	23.49	0.0001	**0.014**
20. Drinking to get out of a bad mood	1.56 (2.39)	1.06 (1.48)	n.d.	27.34	0.0001	**0.016**
**Emotional discharge**	α = 0.82	α = 0.78	α = 0.76	(2, 1787)		
44. Letting my emotions come to the surface by discharging	2.56 (1.58)	3.00 (1.61)	2.40 (1.73)	19.41	0.0001	**0.021**
45. Manifesting my emotion, verbalizing it and expressing it as strongly as possible with my face, with my gestures, with the way I behave	2.31 (1.69)	2.63 (1.69)	2.74 (1.73)	8.91	0.001	0.009
**Confrontation**	α = 0.69	α = 0.78	α = 0.71			
46. Manifest emotion to the person responsible for what happened in order to change things	3.00 (1.60)	4.42 (3.17)	3.24 (1.79)	64.35	0.0001	**0.067**
47. Speaking with sarcasm and irony to the people who provoked the situation	2.25 (1.76)	2.63 (1.75)	2.78 (1.83)	11.34	0.0001	**0.012**
48. Showing my discomfort to the people who provoked the situation by behaving with indifference towards them	2.47 (1.68)	2.63 (1.71)	2.74 (1.73)	2.49	0.083	0.003

Note. n.d. = no data available in this study. In bold the eta squares that explain 1 percent or more of the variance.

**Table 6 ijerph-20-06865-t006:** Relationship between forms of regulation, modification of the situation and social relations, with adjustment and well-being.

	Adjust.	Adjust.	Adjust.	WB1	WB2	WB3
	E1	E2	E3	E1	E2	E3
Direct coping #	0.21 **	0.35 **	0.31 **	0.23 **	0.32 **	0.39 **
Social support #	0.09 **	0.19 **	0.29 **	0.14 **	0.18 *	0.41 *
*Abandon #*	−0.23 **	−0.31 **	0.01	−0.44 **	−0.40 **	−0.06
Altruism	0.08 *	0.07 *	n.d.	0.05	0.05	n.d.
Mediation	−0.07 *	0.004	n.d	−0.06	−0.003	n.d.
Negociation	0.21 **	0.21 **	n.d.	0.19 **	0.18 **	n.d.
Delegatión	−0.11 **	−0.05	n.d.	−0.09 **	−0.08 **	n.d.
Rituals #	−0.06	0.05	0.14 *	−0.21 **	0.10	0.12 *
Social Participation	0.05	0.06 *	n.d.	−0.09 *	0.06 *	n.d.

Note. n.d. = no data * *p* < 0.05; ** *p* < 0.01. # Total scores for the two or more items of each strategy as described in Table 2 and Table 3. Adjust. = Adjustment score or attaintment of adaptive goals in work related emotional/stressful episodes. WB = Wellbeing measured by scale PHI. Dysfunctional or non-adaptive strategies in italics.

**Table 7 ijerph-20-06865-t007:** Association between forms of regulation: orientation of attention and cognitive change, with adjustment and well-being.

	Adjust.1	Adjust.2	Adjust.3	WB 1	WB 2	WB 3
Distraction #	0.05	0.14 **	0.27 **	0.005	0.15 *	0.29 **
Acceptance #	0.06	0.03	0.33 **	0.02	0.08 *	0.51 **
Gratitude self-reward #	0.19 **	0.12 *	0.38 **	0.17 **	0.17 **	0.52 **
Spiritual #	0.002	−0.05	n.d.	−0.02	0.06	n.d.
*Rumination #*	−0.04	0.04	0.32 **	−0.15 **	−0.06 *	0.31 **
Reevaluation #	0.26 **	0.25 **	0.30 **	0.25 **	0.26 **	0.30 **
*Social Comparison #*	−0.07 *	−0.02	0.30 **	−0.17 **	−0.08 *	0.30 **
Search for information	0.17 **	0.22 **	n.d.	0.05	0.11 **	n.d.
ERQreevaluation	0.27 **	0.17 **	n.d.	0.32 **	0.30 **	n.d.

Note. n.d. = no data * *p* < 0.05; ** *p* < 0.01. # Total scores for the two or more items of each strategy as described in Table 4. Adjust = Adjustment score or attainment of adaptive goals in work related emotional/stressful episodes. WB = Wellbeing measured by scale PHI. Dysfunctional or non-adaptive strategies in italics.

**Table 8 ijerph-20-06865-t008:** Association between forms of regulation: modulation of response, with adjustment and well-being.

	Adjust.1	Adjust.2	Adjust.3	WB1	WB2	WB3
Suppression #	−0.19 **	−0.16 **	0.04	−0.36 **	−0.21 **	0.12
Active physiological regulation #	0.21 **	0.12 *	0.38 **	0.18 **	0.11 **	0.35 **
*Passive physiological regulation #*	−0.09 *	−0.07 *	n.d.	−0.27 **	−0.10 **	n.d.
Use Humor affection#	0.09 *	0.13 **	−0.03	0.08 *	0.13 **	−0.11
Regulated emotional expression#	0.10 **	0.16 **	0.36 **	−0.03	0.07 *	0.40 **
*Discharge Venting #*	−0.06	−0.03	0.02	−0.07	0.083 *	−0.07
*Confrontation #*	−0.17 **	−0.07 *	0.02	−0.29 **	−0.04	−0.07
*ERQsuppression*	−0.04	−0.12 **	n.d.	−0.20 **	−24 **	n.d.
EROSden	n.d.	0.21 **	0.03	n.d	0.24 **	−0.07
EROSmen	n.d	0.10 **	0.04	n.d	0.11 **	0.12

Note. n.d. = no data. * *p* < 0.05; ** *p* < 0.01. # Total scores for the two or more items of each strategy as described in Table 5. Adjust. = Adjustment score or attainment of adaptive goals in work related emotional/stressful episodes. WB = Wellbeing measured by scale PHI. EROSden = eros enhancement others emotions, EROSmen = eros increasing other’s negative emotions. Dysfunctional or non-adaptive strategies in italics.

**Table 9 ijerph-20-06865-t009:** Association between forms of regulation with creativity and innovation or applied creativity and health on workers (E1) and humor in students (E2).

	E1		E2				
	ECI	CL	HSQ				SF36
			AF	AAF	HA	AAD	
Direct coping	0.17 **	0.09	−0.04	−0.14	0.03	−0.12	0.18 **
Social support	0.23 **	0.14	−0.36	0.25	0.38 *	−0.05	0.04
*Abandon*	0.02	−0.24 *	0.01	−0.30	0.25	0.22	−0.30 *
Distraction	0.13	−0.003	−0.01	0.02	0.46 **	0.22	0.04
Acceptance	0.17 *	0.000	−0.22	0.11	0.32 +	0.02	−0.08 *
Gratitude self-reward	0.15 +	0.12	−0.10	0.21	0.49 **	0.31 +	0.08 *
Spiritual	0.32 **	0.06	0.01	−0.16	0.21	−0.16	−0.06 *
*Rumination*	0.25 **	−0.11	0.02	0.04	0.32	−0.12	−0.09 **
Reevaluation	0.25 **	0.24 *	−0.14	0.35 *	0.37 *	0.38 *	0.10 **
Social comparison	0.29 **	0.02	−0.35 *	0.16	0.21	0.04	−0.15 **
*Supression*	−0.20 *	−0.23 *	−0.11	−0.03	0.02	0.37 **	−0.15 *
Active physiological regulations	0.18 *	0.29 *	−0.14	0.32 +	0.27 +	0.36 *	0.12 **
*Pasive physiological regulations*	0.10	−0.006	0.08	0.19	0.62 **	0.52 **	−0.19 **
Humor affection	0.04	0.15	−0.08	0.24	0.20	0.28 +	0.09 **
Regulated emotional expression	0.29 **	0.21	0.19	0.10	0.11	0.21	−0.02
*Discharge/Venting*	0.44 **	0.007	0.04	0.50 **	0.39 *	0.33 +	−0.02
*Confrontation*	0.06	−0.07	−0.22	0.10	0.37 *	0.38 *	−0.02
Altruism	0.14 +	0.05	−0.06	0.08	0.37 *	0.34 *	−0.08
Mediation	0.07	−0.05	−0.23	0.25	0.11	−0.05	−0.01
Negotiation	0.03	0.14	−0.05	0.17	−0.03	−0.08	0.12 **
*Delegation*	0.16 +	−0.14	−0.05	0.32 +	0.52 **	0.54 **	−0.08
Rituals	0.19 *	0.06	0.01	−0.16	0.22	−0.16	−0.06
Social Participation	0.26 **	0.02	0.10	0.06	0.23	0.14	0.007
Search for information	0.36 **	0.27 *	0.07	0.22	0.19	0.16	0.06 +
ERQreevaluation	0.19 *	−0.02	0.30 +	0.29 +	−0.19	0.12	0.21 **
*ERQsupression*	−21 *	−0.33 **	−0.14	0.008	0.03	0.23	−0.14 *
Adjustment	0.33 *	0.15	−0.20	−0.04	−0.17	−0.04	0.29 **
Wellbeing measured by PHI	n.d.	n.d.	−0.18	0.35 *	−0.15	−0.09	0.60 **
Emotional Creativity Inventory	n.d.	0.29 *	n.d.	n.d.	n.d.	n.d.	n.d.

Note. + *p* < 0.10; * *p* < 0.05; ** *p* < 0.0. Emotional creativity analyzes were performed on *N* = 104 workers and creativity or innovation in its ideation phase in the face of a labor conflict on *N* = 55. The humor style questionnaire (HSQ) was applied to *N* = 25 Chilean university students. E.C = emotional creativity. CL = labor dispute. AF = affiliative, AFF = self-assertive, HA = aggressive, ADD = self-deprecating. n.d. = no data. Dysfunctional or non-adaptive strategies in italics.

## Data Availability

The data are not publicly available because a confidentiality agreement was signed where it is explicitly stated that they will only be used by the persons responsible for the study for didactic or research purposes.

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
