# Peer review of "Emotion Regulation Strategies in Educational, Work and Sport Contexts: An Approach in Five Countries"

_ijerph, 2023, doi:10.3390/ijerph20196865_

Round 1

Reviewer 1 Report

Dear authors,

Congratulations on the work done, it is an issue that can be interesting for readers. However, before being published, it is necessary to make great changes in the manuscript.

You need a complete reformulation. I leave you some comments that I hope will help you

A description of the description of the participants of the participants is necessary, perhaps a table helps to organize the ideas. Also in which each of the samples contributes to the consequence of the objectives.

Has a calculation been made on the size of the necessary maudra to perform the analyzes that have been done in the study? It seems necessary that this contribution in the manuscript to there is an introduction at the same time three study and many statiny analyzes know what mudra each of them is made.

At home, are one of the studies different or the same person can be included in several studies?

The section of methods insufficient.

There is no procedure section that explains that it has been done in the studies, how are the data collected? How is the study structured? Diagrams of some flow can be of great interest to readers because they can clarify that it has been done and in the study and how

There is no data analysis section

The resulting section is a mess, it is complex. You need full reformulation. I believe that this is due to deficiencies in the manuscript as a good apart of methods in which the procedure and analysis of the data are explained.

Author Response

Reviewer 1

1.- A description of the description of the participants is necessary; perhaps a table helps to organize the ideas. Also in which each of the samples contributes to the consequence of the objectives.

It is true. Participants, recruitment methods, year of data collection, response rate and other aspects are described in the new version. See below answer to comments.

2.- Has a calculation been made on the size of the necessary maudra to perform the analyzes that have been done in the study? It seems necessary that this contribution in the manuscript to there is an introduction at the same time three study and many statiny analyzes know what mudra each of them is made.

A paragraph was included in response to this comment - see below.

For a probability of a Type I error of α = .05 and a probability of a Type II error (1 - β) of .80, and for a typical effect size in social psychology of r =.21 (Richard et al, 2003) or of r=.24 (Lovakov & Agadullina, 2021) the required sample size is N=176 and N=134 respectively (Rosenthal & Rosnow, 2008) so that our samples meets the statistical power requirement.

Lovakov, A., & Agadullina, E. (2021). Empirically Derived Guidelines for Interpreting Effect Size in Social Psychology. Eur J Soc Psychol. 51:485–504. https://www.doi.org/10.1002/ejsp.2752

Richard, F. D., Bond, C. F., Jr., & Stokes-Zoota, J. J. (2003). One hundred years of social psychology quantitatively described. Review of General Psychology, 7, 331–363.

Rosenthal, R. & Rosnow, R.L. (2008). Essentials of Behavioral Research: Methods and Data Analysis. Third Edition. New York: McGraw-Hill.

3.- At home, are one of the studies different or the same person can be included in several studies?

The studies are different. The participants are not repeated and each person answered only one questionnaire.

4.- The section of methods insufficient.

There is no procedure section that explains that it has been done in the studies, how are the data collected? How is the study structured? Diagrams of some flow can be of great interest to readers because they can clarify that it has been done and in the study and how

A figure and table describing variables and scales were included, specifying what was done.

The following figure and table present the variables and scales applied in the studies (see Table and Figure 1).

Figure 1. - Study model: dispositional variables (psychological well-being and health-related quality of life and trait emotional creativity), mediating variables (regulation) and criterion variables (adjustment to stress and applied creativity or innovation as a solution to a labor conflict)

First, measures of self- (MARS and HSQ) and heteroregulation (EROS) were applied. Second, measures of stress adjustment and well-being (PHI). MARS and PHI were applied in all three studies. EROS only in students and athletes sample. Third, measures of health related quality of life, trait creativity (ECI-S) and performance were only applied in study one. Fourth, styles of humor scale were applied only in study 2. The design sought to examine the relationship of affective regulation to stress adjustment and well-being, as well as to trait and performance creativity.

Table 1

Variables, scales and studies included in the study

Variable

Scale

Applied in study

Affect regulation scales

Affect self-regulation

MARS strategies

1, 2 and 3 (7 items not included in E3)

ERQ reappraisal and suppression

1 and 2

HSQ Humor styles

2

Affect Heteroregulation

EROS

2 and 3

Criterion and covariates variables

Adjustment to stress

MARS-adjustment

1, 2 and 3

Wellbeing and health

Psychological well-being

PHI

1, 2 and 3

Health related quality of life 

SF-36 

1

Creativity

Trait emotional creativity

ECI-S

1

Emotional creativity performance

Solution to a labor conflict evaluated by judges in creativity

1

5.- There is no data analysis section

A data analysis section was included

Data analysis

All the scales used have been translated and validated in Spain and Latin America, showing reliability and structural validity (da Costa et al., 2014; 2015). They are also formulated in standard international Spanish. In addition, cross-cultural value studies such as the World value survey show the existence of a shared Catholic European and Latin American cultural value domain (Inglehart & Welzel, 2005). This justifies in our opinion to carry out a general analysis. Reliability analyses were carried out for each scale per sample. Means comparison between samples of the use of regulatory strategies was carried out to contrast the exploratory H1. Means of regulation strategies were compared between samples to explore differences between work, study and sport contexts. Analysis comparing how students, workers and athletes regulated emotions using only Spaniards samples shows similar results to general ones. Subsequently, regulation strategies were correlated with stress adjustment and psychological well-being. Correlations between strategies with the variables of adjustment to stress and well-being in all samples allows us to examine H2. In more exploratory analysis, we examine the relationship of affect regulation with creativity in solving a work conflict. In the sample of workers, trait creativity and creativity of work conflict resolution were correlated with regulation strategies. In the student sample, the relationship between humor styles and regulation and criterion variables is examined. For a probability of a Type I error of α = .05 and a probability of a Type II error (1 - β) of .80, and for a typical effect size in social psychology of r =.21 (Richard et al., 2003) or of r = .24 (Lovakov & Agadullina, 2021) the required sample size for correlation analysis is N=176 and N=134 respectively (Rosenthal & Rosnow, 2008) so that our samples meets the statistical power requirement. Finally, the mediational role of strategies between psychological well-being and stress adjustment is carried out to examine H3. A mediational analysis using Process was carried out in the student’s sample. For student large sample basic multiple regression analysis prerequisites (homoscedasticity...) were meet. Psychological well-being as global evaluation of person disposition is arguable the cause (X variable) of strategies of coping (M or mediation variables) as well a cause of adjustment to stress in the achievement domain (Y or effect or dependent variable). Moreover, PWB correlated with both coping strategies and adjustment, and coping with adjustment – meeting four steps that establish mediation X-M-Y are all correlated (see Kenny https://davidakenny.net/cm/mediate.htm). However, recall that Hayes states that these correlations need not be significant in order to apply a mediational analysis. In our case, we only included variables that were significantly correlated in any case. Program was Hayes’s Process. Effect size estimation for mediation was carried out as suggested by a reviewer with Kenny and another computer software. Using Kenny MedPower https://davidakenny.shinyapps.io/MedPower/ a sample of 250 is required for power .80 or higher in the direct and indirect coefficients of a mediational analysis. For a probability of a Type I error of α = .05 and a probability of a Type II error (1 - β) of .80, and for a mediation coefficient of r=.20, and supposing 10 covariables, a sample of N=276 is required – using https://xuqin.shinyapps.io/CausalMediationPowerAnalysis/ so our sample meets the assumptions and statistical power requirement for mediational analysis

6.- The resulting section is a mess, it is complex. You need full reformulation. I believe that this is due to deficiencies in the manuscript as a good apart of methods in which the procedure and analysis of the data are explained

It is true. In the current version, we first present the results comparing the mean use of regulatory strategies to contrast the exploratory H1. Eta square was included and differences are discussed by reference to explained variance. Then we present the correlations between strategies with the variables of adjustment to stress and well-being in all samples, examining H2. In the sample of workers, we examine the relationship with creativity trait and innovation or creativity in solving a work conflict. In the student sample, the relationship between humor styles and the criterion variables is examined. Finally, the mediational role of strategies between psychological well-being and stress adjustment is carried out to examine H3. 

See data analysis section above

File with corrections is attached

Reviewer 2 Report

The article entitled "Emotion Regulation Strategies in Educational, Work and Sport Contexts: an Approach in Five Countries" addresses the important issue of emotion regulation.

However, in the current version it contains irregularities that need to be corrected.

I list my comments below:

1. The methodological part is incomplete; it lacks a lot of important information.

2. Please describe the study in detail: was the approval of the ethics committee obtained in each country and for each study?

3. How were participants recruited in each of the three studies?

4. What was the response/attrition rates?

5. How was the data collected? Traditionally (paper and pencil) or online? If online, how was data collection controlled, e.g. how was it secured against multiple filling in of a form by one person?

6. What were the inclusion/exclusion criteria in each study and how were they checked if the study was conducted online?

7. Please specify in what exact time data was collected for each of the three studies.

8. Data analysis subchapter is missing. Please describe the data analysis plan in it, including an explanation on what basis parametric statistical tests were used.

9. Please describe how the sample and effect size was calculated.

10. What level of statistical significance did the authors use?

11. In general, there is no explanation based on what assumptions the authors decided to study different groups (students, workers, practicing sports) in different countries, but without taking care to study representatives of each of these groups in each country. In addition, they analyze the results collectively without taking into account the cultural context. I understand that all countries are Spanish speaking. However, it is not methodologically correct to analyze all the results together, especially since people practicing sports were examined only in Spain, which is the only European country and the cultural context related to, for example, the regulation of emotions is undoubtedly different there. In such a situation, comparing further i.e. athletes with any working participants is theoretically unjustified.

12. Frankly, the presentation of the study suggests that the different authors conducted completely independent studies with different groups of participants, and when they realized they were using the same measurement tools, they decided to join forces and present their independent studies as a comparative study. I assume that this was not the case, but the authors did not take care to explain the course of the study in detail and thus avoid such suspicions.

13. In lines 242-243 the authors write: “The reliability of the scale in these studies was acceptable α=.71 E1; α=.64 E2” – firstly, there is no information about Cronbach's alpha value for E3, secondly, α=.64 is below the recommended value of .70, which means that the results obtained in a measurement with such low reliability should be treated with caution. A broader (than just noticing) commentary on this situation should certainly be included in the Study limitations section.

14. Similarly, in lines 278-279: “The reliability in improving and worsening emotions was adequate in both studies (α = .76 and .61 E2; α = .75 and .89 E3), the value of α = .61 is not adequate and the results obtained when measuring with a given tool should be commented on.

15. Again: “This instrument consists of 32 items and four dimensions: affiliative humor (α=.40 if item 29 is eliminated), self-enhancing humor (α=.68), aggressive humor (α=.61 if item 23 is eliminated), and self-defeating humor (α=.70)” – results for affiliative humor should not be analyzed at all, and for self-enhancing humor and aggressive humor should be interpreted with caution.

16. The data presented in Figures 2-5 are illegible and inconsistent with the principles of presenting the results of comparative analyses.

17. The methodological assumptions of the mediation analysis should first be described in detail in the Data analysis section (what program was used to carry out the mediation analysis, what method was adopted, was the power for all of the analyzed paths checked? (this can be done using the MedPower calculator by Kenny (2017)

Kenny DA (2017) MedPower: An interactive tool for the estimation of power in tests of mediation [Computer software].

18. The discussion is very extensive and insightful. However, I will not refer to it, because I have doubts about the reliability of the results obtained.

19. In the case of cite two or more works within the same parentheses they should be in alphabetical order of author. Please check the entire text in this regard. In most cases, in the cited sources, the authors do not adhere to this rule in some citation, for example line 146: (Soroa et al., 2015; Kuska et al., 2022), or line 505: (Madjar et al. ., 2002; da Costa & Paez, 2015).

Author Response

Reviewer 2

  1. The methodological part is incomplete; it lacks a lot of important information.

It is true. See above answer 1 to 6 to Reviewer 1.

  1. Please describe the study in detail: was the approval of the ethics committee obtained in each country and for each study?

No. The approval for the global project of the ethics committee was carried out at the universities of the authors at that time (Universidad Autónoma, Chile, Universidad del Pais Vasco, Spain).

  1. How were participants recruited in each of the three studies?

Procedure

For the study with employees, they were recruited by doctoral students and professors of organizational psychology. Part of the questionnaires were collected face-to-face and part online. An encrypted email was sent to the participants who accessed a link to answer the survey. An encrypted email was sent to the participants who accessed a link to answer the survey. They could only reply once to the link that was sent to them - it was not a free access link.

For the student’s study, the self-report paper and pencil scales were applied of line or face to face collectively, after requesting informed consent, in a single session by the researcher’s authors to young volunteers studying human sciences and education in different Spanish speaking institutions, who answered individually by self-applying the questionnaire. Participants were Spanish and Latin-American university students.

For the study with athletes, the self-report paper and pencil instruments were administered collectively, after requesting informed consent from the athletes who participated voluntarily. The participants were asked to answer the questionnaires thinking about the collective sport they most frequently practiced and in which they participated in competitions. Participants were Spanish athletes.

Response time of the scales was 30 to 40 minutes.

  1. What was the response/attrition rates?

For data collected by encrypted mail link response rate was 30%. For face to face data response rate was close to 100% because they were controlled samples.

  1. How was the data collected? Traditionally (paper and pencil) or online? If online, how was data collection controlled, e.g. how was it secured against multiple filling in of a form by one person?

Students and athlete’s data was collected by paper and pencil scales. – see above answer Reviewer 2 number 3.

For the study with employees, they were recruited by doctoral students and professors of organizational psychology. Part of the questionnaires were collected face-to-face and part online. An encrypted email was sent to the participants who accessed a link to answer the survey. They could only reply once to the link that was sent to them - it was not a free access link.

  1. What were the inclusion/exclusion criteria in each study and how were they checked if the study was conducted online?

The inclusion criterion for all samples was to be of legal age and not to have cognitive limitations.

The researchers who sent the links or collected the questionnaires checked the age and cognitive competence of each participant - who was personally known in each case.

  1. Please specify in what exact time data was collected for each of the three studies.

The three studies were developed in parallel during the years 2015-2016 academic years as part of a research project on emotional regulation.

  1. Data analysis subchapter is missing. Please describe the data analysis plan in it, including an explanation on what basis parametric statistical tests were used.

Data analysis

All the scales used have been translated and validated in Spain and Latin America, showing reliability and structural validity (da Costa et al., 2014; 2015). They are also formulated in standard international Spanish. In addition, cross-cultural value studies such as the World value survey show the existence of a shared Catholic European and Latin American cultural value domain (Inglehart & Welzel, 2005). This justifies in our opinion to carry out a general analysis. Reliability analyses were carried out for each scale per sample. Means comparison between samples of the use of regulatory strategies was carried out to contrast the exploratory H1. Means of regulation strategies were compared between samples to explore differences between work, study and sport contexts. Analysis comparing how students, workers and athletes regulated emotions using only Spaniards samples shows similar results to general ones. Subsequently, regulation strategies were correlated with stress adjustment and psychological well-being. Correlations between strategies with the variables of adjustment to stress and well-being in all samples allows us to examine H2. It was also examined the relationship of affect regulation with creativity in solving a work conflict. In the sample of workers, trait creativity and creativity of work conflict resolution were correlated with regulation strategies. In the student sample, the relationship between humor styles and regulation and criterion variables is examined. For a probability of a Type I error of α = .05 and a probability of a Type II error (1 - β) of .80, and for a typical effect size in social psychology of r =.21 (Richard et al., 2003) or of r = .24 (Lovakov & Agadullina, 2021) the required sample size for correlation analysis is N=176 and N=134 respectively (Rosenthal & Rosnow, 2008) so that our samples meets the statistical power requirement. Finally, the mediational role of strategies between psychological well-being and stress adjustment is carried out to examine H3. A mediational analysis using Process was carried out in the student’s sample. For student large sample basic multiple regression analysis prerequisites (homoscedasticity...) were meet. Psychological well-being as global evaluation of person disposition is arguable the cause (X variable) of strategies of coping (M or mediation variables) as well a cause of adjustment to stress in the achievement domain (Y or effect or dependent variable). Moreover, PWB correlated with both coping strategies and adjustment, and coping with adjustment – meeting four steps that establish mediation X-M-Y are all correlated (see Kenny https://davidakenny.net/cm/mediate.htm). However, recall that Hayes states that these correlations need not be significant in order to apply a mediational analysis. In our case, we only included variables that were significantly correlated in any case. Program was Hayes’s Process. Effect size estimation for mediation was carried out as suggested by a reviewer with Kenny and another computer software. Using Kenny MedPower https://davidakenny.shinyapps.io/MedPower/ a sample of 250 is required for power .80 or higher in the direct and indirect coefficients of a mediational analysis. For a probability of a Type I error of α = .05 and a probability of a Type II error (1 - β) of .80, and for a mediation coefficient of r=.20, and supposing 10 covariables, a sample of N=276 is required – using https://xuqin.shinyapps.io/CausalMediationPowerAnalysis/ So our sample meets the assumptions and statistical power requirement for mediational analysis.

Inglehart, R & C. Welzel. (2005). Modernization, Cultural Change and Democracy: The Human Development Sequence. New York: Cambridge University Press.

  1. Please describe how the sample and effect size was calculated.

See Reviewer 1 answer

For a probability of a Type I error of α = .05 and a probability of a Type II error (1 - β) of .80, and for a typical effect size in social psychology of r = .21 (Richard et al, 2003) or of r =.24 (Lovakov & Agadullina, 2021) the required sample size is N =176 and N = 134 respectively (Rosenthal & Rosnow, 2008) so that our samples meets the statistical power requirement.

  1. What level of statistical significance did the authors use?

Statistical significance for Type I error was of α = .05 and for probability of a Type II error (1 - β) of .80

  1. In general, there is no explanation based on what assumptions the authors decided to study different groups (students, workers, practicing sports) in different countries, but without taking care to study representatives of each of these groups in each country. In addition, they analyze the results collectively without taking into account the cultural context. I understand that all countries are Spanish speaking. However, it is not methodologically correct to analyze all the results together, especially since people practicing sports were examined only in Spain, which is the only European country and the cultural context related to, for example, the regulation of emotions is undoubtedly different there. In such a situation, comparing further i.e. athletes with any working participants is theoretically unjustified.

See below justifications

In all studies there were Spanish and Latin American participants. The shared objective was to compare how students, workers and athletes regulated emotions in the common domain of achievement or performance. In the presentation of the instruments, it was emphasized that we were asking how people managed emotions in the non-interpersonal and non-family environment, but in the area of performance or achievement, i.e., studies, work and competitive sports.

Analysis comparing how students, workers and athletes regulated emotions using only Spaniards samples shows similar results to general ones.

 A clear limitation is that the samples, although massive, were convenience and not random - this is discussed in limitations.

All the scales used have been translated and validated in Spain and Latin America, showing reliability and structural validity. They are also formulated in standard international Spanish. In addition, cross-cultural value studies such as the world value survey show the existence of a shared Latin European and Latin American cultural value domain. This justifies in our opinion to carry out a general analysis.

For the student’s study, the self-report paper and pencil scales were applied of line or face to face collectively, after requesting informed consent, in a single session by the researcher’s authors to young volunteers studying human sciences and education in different Spanish speaking institutions, who answered individually by self-applying the questionnaire. Participants were Spanish and Latin-American university students.

For the study with employees, they were recruited by doctoral students and professors of organizational psychology. Part of the questionnaires were collected face-to-face and part online. An encrypted email was sent to the participants who accessed a link to answer the survey. Participants were Spanish and Latin-American workers.

For the study with athletes, the self-report paper and pencil instruments were administered collectively, after requesting informed consent from the athletes who participated voluntarily. The participants were asked to answer the questionnaires thinking about the collective sport they most frequently practiced and in which they participated in competitions. Participants were Spanish athletes.

  1. In lines 242-243 the authors write: “The reliability of the scale in these studies was acceptable α=.71 E1; α=.64 E2” – firstly, there is no information about Cronbach's alpha value for E3, secondly, α=.64 is below the recommended value of .70, which means that the results obtained in a measurement with such low reliability should be treated with caution. A broader (than just noticing) commentary on this situation should certainly be included in the Study limitations section.

Okay. This paragraph was deleted and alpha were reported in Tables. Scales with low reliabilities were deleted in results and were included as limitations in discussion.

  1. Similarly, in lines 278-279: “The reliability in improving and worsening emotions was adequate in both studies (α = .76 and .61 E2; α = .75 and .89 E3), the value of α = .61 is not adequate and the results obtained when measuring with a given tool should be commented on.

Okay. Scales with low reliabilities were deleted in results and were included as limitations in discussion.

  1. Again: “This instrument consists of 32 items and four dimensions: affiliative humor (α=.40 if item 29 is eliminated), self-enhancing humor (α=.68), aggressive humor (α=.61 if item 23 is eliminated), and self-defeating humor (α=.70)” – results for affiliative humor should not be analyzed at all, and for self-enhancing humor and aggressive humor should be interpreted with caution.

Okay results for affiliative humor were deleted

As study limitations, it is noted that this is a cross-sectional study that prevents making causal statements, that the samples were large but convenience samples, and that the reliabilities of some of the scales, like EROs worsening heteroregulation scale and affiliative humor, - in some of the samples in this study - are low. Regarding the results of humor and creativity, the small samples do not allow for generalizing the findings, however, they are consistent with previous studies. Future studies need to consider these limitations when comparing different nations and exploring whether regulation strategies are influenced by cultural patterns

  1. The data presented in Figures 2-5 are illegible and inconsistent with the principles of presenting the results of comparative analyses.

Data was described in standard mean tables. F, signification and eta square are reported.

We will discuss significant mean differences that explain at least 1% of the variance or an eta squared of .01 which is equivalent to a small effect size of r=.10 or d=.20.

Table 2

Reliabilities (a) of the dimensions, mean, standard deviation (SD) and one-way analysis of variance of situation modification and social ties items

Measure

Study 1 workers

Study 2 students

Study 3 Athletes

F (2,1787)

p

η²

Direct problem solving and planning. Instrumental coping

α= 76

α= 73

α= 71

04. Develop a plan to deal with what happened and be able to do something to change the situation

3.80 (1.54)

4.04 (1.45)

4.18 (1.53)

6.9

0.001

.007

05. Act or do something to solve the problem that caused my mood

4.10 (1.39)

4.24 (1.25)

4.78 (1.39)

15.94

0.0001

.02

(1, 1644)

06. Make plans or make a decision to avoid military problems in the future

4.14 (1.32)

4.32 (1.28)

n.d

7.71

0.0056

.005

Seeking emotional support

α= 92

α= 80

n.d

52. Talking to someone about how I feel.

3.84 (1.70)

4.10 (1.64)

n.d

9.76

0.018

.006

53. Speak up for understanding and support

3.52 (1.78)

3.91 (1.71)

n.d

19.99

0.0001

.012

Search for instrumental and information support

α= 88

(2,1787)

54. Asking someone else for help in resolving the situation that triggered my mood

3.10 (1.72)

3.31 (1.78)

3.40 (1.77)

3.54

0.03

.004

55. Talking to someone who could give me advice and guidance

3.61 (1.66)

3.84 (1.67)

3.50 (1.92)

4.88

0.007

.005

56. Ask someone who faced a similar problem what they did

3.23 (1.71)

3.34 (1.17)

4.01 (1.81)

17.16

0.0001

.019

Behavioral avoidance and withdrawal

α= 72

α= 69

α= 67

07. Leaving or leaving the situation

1.63 (1.46)

1.70 (1.38)

1.89 (1.60)

2.04

0.13

.002

08. Acting as if nothing is wrong

1.62 (1.42)

1.88 (1.53)

1.71 (1.53)

6.22

0.002

.007

09. Giving up or doing nothing: not trying to control the situation

1.15 (1.25)

1.21 (1.29)

1.15 (1.21)

0.50

0.60

.0001

(1, 1644)

13. Avoid contact with people associated with the problem

2.74 (1.72)

2.71 (1.70)

n.d

3.96

0.046

.002

14. Seeking to be alone

2.30 (1.76)

2.63 (1.70)

n.d

14.64

0.0001

.009

35. Try to accept my fate, which is inevitable, it has to be this way

2.33 (1.86)

2.70 (1.80)

n.d

16.45

0.0001

.001

Note. n.d- = no data available in this study

Table 3

Reliabilities (a) of the dimensions, mean, standard deviation (SD) and one-way analysis of variance of strategies measured with a single item in the Modification of Situation and social relationships facet

Measure

Study 1 workers

Study 2 students

Study 3 Athletes

F (1, 1644)

p

η²

Altruism

41. Forget about my situation and help someone else

2.37 (1.53)

2.45 (1.49)

n.d

1.13

0.28

.001

Mediation

57. Ask someone to mediate or intervene in relation to what happened

1.55 (1.50)

1.64 (1.54)

n.d

1.37

0.24

.001

Negotiation

58. Talk about what happened with the people involved to negotiate or reach an agreement

2.92 (1.68)

3.28 (1.63)

n.d

19.03

0.0001

.011

Rituals

(2,1787)

Private

59. Participate in or organize a private ceremony (I wrote an e-mail about what happened, reorganized photos, letters)

1.00 (1.42)

0.91 (1.41)

4.09 (1.48)

325.24

0.0001

.27

Public

60. Participate on or organize a public ceremony (demonstration, mass, commemoration)

0.62 (1.25)

0.65 (1.28)

1.89 (2.08)

56.99

0.0001

.06

Social Participation

(1, 1644)

62. Get involved in political or social activities

1.24 (1.53)

1 (1.55)

n.d

9.7

0.002

0.006

Delegation

64. Putting myself in the hands of others to solve my problem or improve the situation

1.12 (1.32)

1.32 (1.44)

n.d

8.27

0.004

0.005

Note. n.d.= no data available in this study

Table 4

Reliabilities (a) of the dimensions, mean, standard deviation (SD) and one-way analysis of variance of attention orientation and cognitive change items

Measure

Study 1 workers

Study 2 students

Study 3 Athletes

F (2,1787)

p

η²

Distraction

α= 81

α= 78

α= 62

21. Doing something fun, something I really like and enjoy

4.07 (1.48)

4.28 (1.40)

3.66 (2.30)

11.92

0.0001

.013

(1, 1644)

22. Watch TV, read a book, listen to music, etc., to distract myself

4.07 (1.55)

4.35 (1.43)

n.d

14.3

0.0002

.009

(2,1787)

23. Working on something to keep busy to take my mind off my mood

3.73 (1.52)

3.75 (1.54)

3.82 (2.21)

0.19

0.83

.0001

24. Thinking about something else to distract me from my feelings

3.47 (1.48)

2.37 (1.48)

4.73 (1.51)

208.85

0.0001

.19

(1, 1644)

25. Being with people, talking, to forget my mood

3.60 (1.57)

3.83 (1.56)

n.d

8.85

0.003

.005

Gratitude and self-reward

α= 68

α= 71

α= 82

(2,1787)

28. Do something special to reward myself and feel better

2.97 (1.71)

3.03 (1.68)

4.89 (1.39)

84.58

0.0001

.086

30. Trying to think about those things that I’m doing well at

3.60 (1.60)

3.34 (1.51)

4.69 (1.51)

48.41

0.0001

.05

31. Trying to be grateful for the things that are going well in life

3.95 (1.63)

3.84 (1.68)

4.83 (1.88)

21.82

0.0001

.024

Spiritual activity

α= 88

α= 82

n.d

36. To pray, to put my faith in God, to lean on religious things

2.20 (2.19)

1.59 (1.97)

3.32 (2.09)

51.18

0.0001

.054

(1, 1644)

40. Read or do something religious, spiritual

1.87 (2.03)

1.07 (1.64)

n.d

77.94

0.0001

.045

Rumination

α= 64

α= 57

α= 76

(2,1787)

01.Think about how you could have done things differently

3.88 (1.41)

4.09 (1.29)

4.28 (1.51)

7.58

0.0005

.008

02. Trying to understand my feelings by thinking about and analyzing them

4.04 (1.37)

3.97 (1.48)

4.17 (1.57)

1.39

0.24

.0015

03. Repeatedly thinking about what happened, about the emotional effects of the situation

3.77 (1.46)

3.95 (1.48)

4.00 (1.60)

3.42

0.03

.004

Re-evaluation

α= 81

α= 80

α= 77

37. Try to reinterpret the situation, to find a different meaning or sense

3.40 (1.54)

3.31 (1.54)

3.91 (1.67)

9.38

0.0001

.01

38. Trying to see things from a broader perspective

4.04 (1.37)

3.78 (1.35)

4.41 (1.51)

16.77

0.0001

.018

39. Trying to find somethings good in the situation

3.87 (1.49)

3.76 (1.51)

4.00 (1.60)

2.16

0.11

.002

Social Comparison

α= 69

α= 60

α= 59

42.Comparing myself to people who are worse off than I am.

3.00 (2.78)

2.42 (1.71)

3.45 (1.91)

22.47

0.0001

.025

43. Comparing myself to a person with more a means, personal resources and who had done better than me. To take him/her as a model to improve my situation

2.39 (2.58)

1.94 (1.62)

4.40 (1.87)

90.13

0.0001

.092

Information search

(1, 1644)

61. Inform me about my problem or about the situation in order to overcome it or do better

2.93 (1.73)

2.78 (1.75)

n.d

2.97

.085

.002

Note.

Table 5

Reliabilities (a) of the dimensions, mean, standard deviation (SD) and one-way analysis of variance of attention orientation and cognitive change items

Measure

Study 1 workers

Study 2 students

Study 3 Athletes

F (2,1787)

p

η²

Inhibition/suppression

α= 77

α= 66

α= 64

10. Try not to think about what happened, ignore negative emotions

2.44 (1.68)

2.16 (1.44)

2.03 (1.54)

8.27

0.003

.009

11. Try not to show my feelings, to suppress any expression of them

2.64 (1.72)

2.32 (1.68)

2.15 (1.59)

9.33

0.0001

.01

12. Faking or expressing emotion opposite to what you feel

2.24 (1.73)

1.89 (1.60)

1.81 (1.50)

10.33

0.0001

.011

Physiological regulation

Active

α= 69

α= 45

       α= 63

15. Exercise, sport

3.02 (1.95)

2.62 (1.95)

4.40 (1.35)

56.06

0.0001

.059

16. Practicing relaxation, meditation

2.61 (3.05)

1.80 (1.71)

3.58 (1.45)

50.68

0.0001

.054

Passive

α= 79

α= 63

n.d

(1, 1644)

17. Sleeping or napping

3.35 (3.15)

2.76 (1.83)

n.d

22.87

0.0001

.014

18. Eating something to overcome my mood

3.24 (3.19)

2.61 (1.82)

n.d

25.7

0.0001

.015

19. Drinking coffee, caffeinated beverages or tea

2.36 (3.08)

1.77 (1.84)

n.d

23.49

0.0001

.014

20. Drinking to get out of a bad mood

1.56 (2.39)

1.06 (1.48)

n.d

27.34

0.0001

.016

Emotional discharge

α= 82

α= 78

       α= 76

(2,1787)

44. Letting my emotions come to the surface by discharging

2.56 (1.58)

3.00 (1.61)

2.40 (1.73)

19.41

0.0001

.021

45. Manifesting my emotion, verbalizing it and expressing it as strongly as possible with my face, with my gestures, with the way I behave

2.31 (1.69)

2.63 (1.69)

2.74 (1.73)

8.91

0.001

.009

Confrontation

α=69

α= 78

       α= 71

46. Manifest emotion to the person responsible for what happened in order to change things

3.00 (1.60)

4.42 (3.17)

3.24 (1.79)

64.35

0.0001

.067

47. Speaking with sarcasm and irony to the people who provoked the situation

2.25 (1.76)

2.63 (1.75)

2.78 (1.83)

11.34

0.0001

.012

48. Showing my discomfort to the people who provoked the situation by behaving with indifference towards them

2.47 (1.68)

2.63 (1.71)

2.74 (1.73)

2.49

0.083

.003

Note. n.d = no data available in this study

  1. The methodological assumptions of the mediation analysis should first be described in detail in the Data analysis section (what program was used to carry out the mediation analysis, what method was adopted, was the power for all of the analyzed paths checked? (this can be done using the MedPower calculator by Kenny (2017)

Kenny DA (2017) MedPower: An interactive tool for the estimation of power in tests of mediation [Computer software].

For student’s large sample basic assumptions were meet. Program was Process. Effect size estimation for mediation was carried out. See below new paragraph of data analysis – exposed above but repeated here to answer this comment

A mediational analysis using Process was carried out in the student sample. For student large sample basic multiple regression analysis prerequisites (homoscedasticity...) were meet. Psychological well-being as global evaluation of person disposition is arguable the cause (X variable) of strategies of coping (M or mediation variables) as well a cause of adjustment to stress in the achievement domain (Y or effect or dependent variable). Moreover, PWB correlated with both coping strategies and adjustment, and coping with adjustment – meeting four steps that establish mediation X-M-Y are all correlated (see Kenny https://davidakenny.net/cm/mediate.htm). However, recall that Hayes states that these correlations need not be significant in order to apply a mediational analysis. In our case, we only included variables that were significantly correlated in any case. Program was Hayes’s Process. Effect size estimation for mediation was carried out as suggested by a reviewer with Kenny and another computer software. Using Kenny MedPower https://davidakenny.shinyapps.io/MedPower/ a sample of 250 is required for power .80 or higher in the direct and indirect coefficients of a mediational analysis. For a probability of a Type I error of α = .05 and a probability of a Type II error (1 - β) of .80, and for a mediation coefficient of r=.20, and supposing 10 covariables, a sample of N=276 is required – using https://xuqin.shinyapps.io/CausalMediationPowerAnalysis/. So our sample meets the assumptions and statistical power requirement for mediational analysis.

  1. The discussion is very extensive and insightful. However, I will not refer to it, because I have doubts about the reliability of the results obtained.

Limitations of reliabilities are discussed

  1. In the case of cite two or more works within the same parentheses they should be in alphabetical order of author. Please check the entire text in this regard. In most cases, in the cited sources, the authors do not adhere to this rule in some citation, for example line 146: (Soroa et al., 2015; Kuska et al., 2022), or line 505: (Madjar et al.., 2002; da Costa & Paez, 2015).

Done

Round 2

Reviewer 2 Report

I accept the changes provider by Authors.